# Safety Depth in Large Language Models:
# A Markov Chain Perspective

**Ching-Chia Kao**[1,2]*, **Chia-Mu Yu**[3], **Chun-Shien Lu**[1], **Chu-Song Chen**[2]

[1]Institute of Information Science & Research Center for Information Technology Innovation, Academia Sinica
[2]Department of Computer Science and Information Engineering, National Taiwan University
[3]Department of Electronics and Electrical Engineering, National Yang Ming Chiao Tung University

## Abstract

Large Language Models (LLMs) are increasingly adopted in high-stakes scenarios, yet their safety mechanisms often remain fragile. Simple jailbreak prompts or even benign fine-tuning can bypass internal safeguards, underscoring the need to understand the failure modes of current safety strategies. Recent findings suggest that vulnerabilities emerge when alignment is confined to only the initial output tokens. To address this, we introduce the notion of *safety depth*, a designated output position where the model refuses to generate harmful content. While deeper alignment appears promising, identifying the optimal safety depth remains an open and underexplored challenge.

We leverage the equivalence between autoregressive language models and Markov chains to derive the first theoretical result on identifying the optimal safety depth. To reach this safety depth effectively, we propose a cyclic group augmentation strategy that improves safety scores across six LLMs. In addition, we uncover a critical interaction between safety depth and ensemble width, demonstrating that larger ensembles can offset shallower alignments. These results suggest that test-time computation, often overlooked in safety alignment, can play a key role. Our approach provides actionable insights for building safer LLMs.

## 1 Introduction

Although Large Language Models (LLMs) [Touvron et al., 2023, Achiam et al., 2023, Team et al., 2023, Duan et al., 2023, Ouyang et al., 2022] excel in diverse applications, they often produce harmful content. Reinforcement Learning from Human Feedback (RLHF) [Ouyang et al., 2022, Bai et al., 2022] and its variants, Direct Preference Optimization (DPO) [Rafailov et al., 2024] and Kahneman-Tversky Optimization (KTO) [Ethayarajh et al., 2024], aim to mitigate this issue. However, recent studies have shown that adversarially optimized inputs can still elicit harmful content [Qi et al., 2023, Carlini et al., 2024, Chao et al., 2023, Andriushchenko et al., 2024], and even benign fine-tuning can break existing alignments [Qi et al., 2024, Zhan et al., 2024].

A recent study [Qi et al., 2025] uncovered that this vulnerability stems from limiting safety alignment to only the initial output tokens, a practice termed *shallow safety alignment*. They introduced a data augmentation method to deepen alignment, leading to the question: "**How extensive should safety alignment be?**" To address this, we divide the problem into manageable steps. First, we ask, "**What does it mean for an alignment to be safe?**" We consider a finite set $Y$ of harmful content, such as explicit sexual, violent, or private information, that we intend the model to avoid. An LLM is safely aligned if the probability of generating any content in $Y$ is extremely low.

Second, rather than always placing a refusal at the beginning, Qi et al. [2025] randomly picked a *safety depth* from a uniform distribution and insert a refusal response along with harmful instructions

---

*Correspondence to: `d11922015@csie.ntu.edu.tw`

at that position. From a group-theoretic perspective, these insertions represent specific instances of broader transformations on the dataset. Motivated by this, we pose another question: "**What if the dataset is augmented by rotation, forming a cyclic group?**" (see Figure 1). This framework links safety alignment to permutation groups, a topic we explore in Section 4. For readers less familiar with group theory, Appendix A offers a concise overview.

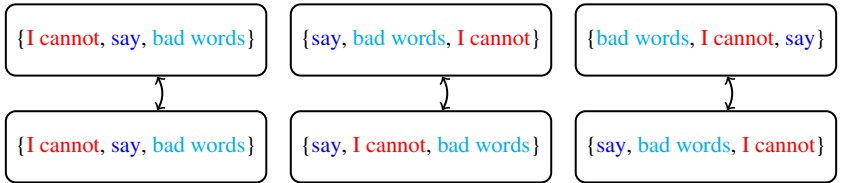

Figure 1: Permutations of phrases used for data augmentation. The top row represents a cyclic group, while the bottom row, as proposed by [Qi et al., 2025], is non-cyclic.

Lastly, while many works [Malladi et al., 2023, Jang et al., 2024, Tomihari and Sato, 2024] rely on the Neural Tangent Kernel (NTK) [Jacot et al., 2018] to analyze fine-tuning and Gerken and Kessel [2024] demonstrates that deep ensembles can become fully equivariant through data augmentation over all group actions, we adopt a Markov chain perspective. Markov chain benefits without relying on the linearization assumptions. Detailed comparisons are discussed in Appendix B.

Our contributions are threefold:

- **Safety Depth.** We formalize the notion of Safety Depth, a designated output position where the model refuses to generate harmful content. Viewing this through a Markov chain lens in Theorem 4.6 provides theoretical assurances of arbitrarily safe behavior via iterative fine-tuning on autoregressive LLMs.

- **Cyclic Augmentation.** Proposition 4.9 establishes that safety guarantees remain intact under cyclic augmentation, while also accelerating convergence. Empirically, this results in consistent improvements in safety scores across multiple LLMs.

- **Ensemble Safety Depth.** Proposition 4.11 shows that ensembles can preserve safety guarantees without per-model training. This provides a practical alternative for users seeking an off-the-shelf solution, though at the expense of longer test-time inference.

## 2 Related Works

**Safety Alignment.** Safety alignment ensures that LLMs adhere to human values, reducing their susceptibility to malicious instructions [Yi et al., 2024]. Li et al. [2025] identified particular "safety layers" that differentiate malicious from normal queries, revealing distinct behaviors when models process adversarial versus benign prompts. Common alignment methods include RLHF [Ouyang et al., 2022, Bai et al., 2022] and DPO [Rafailov et al., 2024], but researchers have also explored alternatives. For instance, Safety Arithmetic [Hazra et al., 2024] is a training-free technique employing parameter arithmetic to suppress harmful outputs while promoting safer ones, and SAFEPATCHING [Zhao et al., 2024, Kim et al., 2024] refines alignment by selectively adjusting model parameters. Another training-free approach [Zhou et al., 2024] can even reverse an LLM's safety alignment. [Zhao et al., 2025, Haldar et al., 2025, Pan et al., 2025, Zhang et al., 2025] also represent the other effort.

**Markov chains and LLMs.** While they may seem unrelated, Markov chains and LLMs share a core principle. Autoregressive LLMs can be viewed as Markov chains with a large but finite state space, and their token-by-token generation mirrors the "memorylessness" property of Markov processes. Zekri et al. [2024] formally demonstrated that an LLM with vocabulary size $D$ and context length $K$ can be represented by a Markov chain of size $O(D^K)$, offering a theoretical lens for studying convergence and generalization properties in LLMs.

**Group theory and LLMs.** Group theory focuses on symmetry, manifesting in phenomena from crystal structures to fundamental forces[2]. In the LLM context, it has been used to test algebraic properties—such as closure, identity, inverse, and associativity—revealing that LLMs often fail to maintain these properties under various testing regimes [Imani and Palangi, 2024]. For instance, LLMs may produce skewed outputs or show abrupt performance drops beyond certain sequence

---

[2]This is distinct from Group Relative Policy Optimization (GRPO) [Shao et al., 2024] by [Guo et al., 2025].

lengths. Conversely, Chang et al. [2025] explored a more constructive angle, illustrating how carefully curated training data can help LLMs learn and uphold algebraic structures more reliably.

# 3 Preliminaries

This section gives an overview of sufficient elements to understand our main theorem, including the Markov chain training procedure, the autoregressive language model as a Markov chain, and group action on the training procedure. We also present a notation table in Table 4 in Appendix C.

**Markov Chain.** Consider a discrete-time Markov chain with $n$ states labeled $1, 2, \ldots, n$. Let $Q_t \in \mathbb{R}^{n \times n}$ denote the transition matrix at time $t$. We assume an *initial* transition matrix $Q_0$ in the context of safety alignment and introduce a small learning rate $\alpha$. The bias is encoded by a matrix $B \in \mathbb{R}^{n \times n}$, where $B$ represents how the bias modifies specific entries of $Q_0$. The bias $B$ can be seen as a defender preference for refusal or an attacker preference for uncensored words that is discounted by a factor $\gamma \in (0, 1)$ at each time $t$. Hence, the transition matrix at time $t$ is given by:

$$Q_t \; = \; Q_0 \; + \; \alpha \, \gamma^t \, B. \tag{1}$$

We require that $Q_t$ remains a valid stochastic matrix; in particular, each row of $Q_t$ must sum to 1, and entries must remain nonnegative. This typically imposes constraints on the magnitude of $\alpha$ and the structure of $B$. The asymptotic behavior of Markov chain is left to Appendix D.

**Autoregressive LLM.** We formally introduce a typical autoregressive LLM following [Zekri et al., 2024]; the detailed inner structure is left to Appendix E. Let $\mathcal{V}$ be a dictionary of size $D$. For context window $K$, define $\mathcal{V}_K^* := \{v \in \mathcal{V}^* : |v| \leq K\}$, which represents a restriction of Kleene closure of $\mathcal{V}$. Consider an autoregressive LLM, $\pi_\theta^{D,K} : \mathcal{V}_K^* \to \Delta(\mathcal{V})$, where $\Delta(\mathcal{V})$ denotes the probability simplex over $\mathcal{V}$ and weights are parameterized by $\theta$. We will drop the superscript $D, K$ when they are of no importance.

Specifically, at inference time we can let $\mathcal{X} \subseteq \mathcal{V}_K^*$ be the set of input documents (token sequences) and $\mathcal{Y} \subseteq \mathcal{V}_K^*$ be the set of output documents. Given an input $x$, the model's output distribution is denoted by $\pi_\theta(\cdot|x) \in \Delta(\mathcal{Y})$, where $\Delta(\mathcal{Y})$ is the set of probability distributions over $\mathcal{Y}$. We write $y \sim \pi_\theta(\cdot|x)$ to denote the sampling output $y$ of this distribution.

From the result of [Zekri et al., 2024], we know that a Markov chain can represent any autoregressive model equivalently. Hence, we have the following assumption that characterizes fine-tuning LLM.

**Assumption 3.1.** Fine-tuning LLM is equivalent to an iterative update of the transition matrix $Q_t$.

**Justification of Assumption 3.1.** This assumption, also used by [Zekri et al., 2024], treats autoregressive LLMs as Markov chains, where fine-tuning corresponds to iterative updates of the transition matrix. While LLMs operate in a high-dimensional parameter space, the observable effect of fine-tuning is to alter token transition probabilities, which closely mirrors transition dynamics in Markov chains. To illustrate, consider a toy model with tokens "I," "cannot," and "comply." Fine-tuning to reinforce the refusal phrase "I cannot comply" increases the probability of the transitions I $\rightarrow$ cannot and cannot $\rightarrow$ comply, which can be modeled as structured updates to a stochastic matrix. This abstraction enables tractable analysis while retaining fidelity to LLM behavior at the token level.

**Data Augmentation via Group Actions.** We introduce the data augmentation via group action and typically leverage this property to analyze the safety alignment for this specially designed dataset. Given a training set $\mathcal{T} = \{(x_i, y_i)\}_{i=1}^N$, we can augment it using group actions.

**Definition 3.2** (Augmented Training Set). The group-augmented training set is defined as:

$$\mathcal{T}_{\text{aug}} = \{(\rho_X(g)x, \rho_Y(g)y) \mid g \in G, (x, y) \in \mathcal{T}\}, \tag{2}$$

where $\rho_X$ and $\rho_Y$ are group representations as illustrated in Appendix A.6.

**Proposition 3.3.** *For a finite group $G$, its action on the training set can be represented as a permutation $\sigma_g$:*

$$\rho_X(g)x_i = x_{\sigma_g(i)} \quad and \quad \rho_Y(g)y_i = y_{\sigma_g(i)}.$$

This leads to the following properties: the size of the augmented training set scales with the group order: $|\mathcal{T}_{\text{aug}}| = |G| \cdot |\mathcal{T}|$, and the augmentation preserves the relationship between inputs and outputs.

**Assumption 3.4.** Fine-tuning LLM on $\mathcal{T}_{\text{aug}}$ is equivalent to matrix conjugate on a bias matrix.

**Justifications of Assumption 3.4.** This assumption states that fine-tuning on cyclically augmented data corresponds to conjugating the bias matrix: $B^{(t)} = P^t B P^{-t}$, where $P$ is a permutation matrix. The augmentation rotates refusal phrases across positions (e.g., "I cannot say bad words" → "say bad words I cannot"), forming a cyclic group of training examples. This ensures that each refusal position receives consistent supervision, and the effect on learning can be captured as a sequence of structured bias transformations. Our theoretical results (e.g., Proposition 4.9 ) show that under this setup, the model converges to a $\delta$-absorbing state at multiple positions, approximating an oracle with full positional coverage.

**Group action on a Markov chain.** Suppose $B^{(t)}$ is obtained by applying a *cyclic group* action to some base matrix $B$ at time $t$. For instance, if $P$ is a fixed permutation of length $n$, then

$$B^{(t)} = P^t B P^{-t}. \tag{3}$$

Typically, $B$ itself may or may not be stochastic, but it is *bounded* in some norm: $\|B^{(t)}\| = \|B\|$ for any matrix norm invariant under permutation. Since $Q_t - Q_0 = \alpha \gamma^t B^{(t)}$, we typically get

$$\|Q_t - Q_0\|_\infty \leq \alpha \gamma^t \|B^{(t)}\|_\infty = \alpha \gamma^t \|B\|_\infty. \tag{4}$$

**Remark 3.5.** *Both permutations, as shown in Figure 1, are bounded due to Eqs. (3) and (4). However, the data augmentation in [Qi et al., 2025] is not a group, which made the size of the augmented dataset hard to control. Moreover, although both data augmentations in [Qi et al., 2025] and our cyclic augmentation are counterfactual, the model's utility remains stable, as shown in Appendix J.*

## 4 Main Result

In this section, we first formally define the safety alignment of autoregressive models to which many LLMs belong. Zekri et al. [2024] has shown that an autoregressive language model can be reinterpreted as a Markov chain over its output space, providing a rigorous framework for analyzing model behavior. Most importantly, we are particularly interested in **safety depth**, a specific position of output responses in the training samples where the model declines to generate potentially harmful content. To ensure the safety of such models, it is desirable that once the model enters a **safety depth**, also called a **refusal state**, it never transitions to generating harmful content. Theorem 4.6 and Corollary 4.8 demonstrate that the probability of leaving refusal states can be made arbitrarily small through iterative fine-tuning. Specifically, repeated minor adjustments that increase the probability of remaining in refusal states will eventually make these states effectively absorbed, providing theoretical guarantees for the model's safety properties under the Markov chain interpretation.

**Definition 4.1** (Safety Alignment). Let a language model $\pi_\theta(\cdot \mid x)$ be a conditional distribution over the set of all possible output sequences $\mathcal{Y}$. Let $Y \subset \mathcal{Y}$ be a subset of "harmful contents." We say the language model is *safely aligned* if:

$$\sup_{x \in \mathcal{X}} \pi_\theta\big(Y \mid x\big) \leq \varepsilon, \tag{5}$$

where $\varepsilon$ is a small threshold. Equivalently, for all $x \in \mathcal{X}$:

$$\pi_\theta\big(Y \mid x\big) = \sum_{y \in Y} \pi_\theta(y \mid x) \leq \varepsilon. \tag{6}$$

This ensures that for every input $x$, the probability of generating harmful content is bounded by a small value $\varepsilon$. On the other hand, recall from [Zekri et al., 2024] that any autoregressive model can be equivalently represented by a Markov chain. Another line of work [Emde et al., 2025] uses a likelihood-ratio test with a guardrail model to provide a universal adversarial bound during inference.

**Definition 4.2** (Safe Alignment in Markov View). Let $\pi_\theta$ be considered as a Markov chain on state space $S$ with transition probabilities $\Pr(s_1 \to s_2)$ for $s_1, s_2 \in S$. We first define the set of harmful states $S_Y \subseteq S$, and then define the set that can reach $S_Y$ with positive probability as:

$$S_Y^\perp := \{s \in S \setminus S_Y \mid \exists n \geq 1, \overset{n}{\Pr}(s \to S_Y) > 0\}, \tag{7}$$

where $\overset{n}{\Pr}(s \to S_Y)$ is the probability from $s$ to harmful state $S_Y$ in $n$ steps. Finally, we can define the block transition matrices:

- $Q = [\Pr(s_1 \to s_2)]_{s_1,s_2 \in S_Y^{\perp}}$ for transitions within $S_Y^{\perp}$

- $Q_{harm} = [\Pr(s_1 \to s_2)]_{s_1 \in S_Y^{\perp}, s_2 \in S_Y}$ for transitions from $S_Y^{\perp}$ to $S_Y$

For any initial distribution $\mathbf{p}_0$ over $S_Y^{\perp}$, the hitting probability for harmful states is:

$$\mathbb{P}(\text{hit } S_Y \mid \mathbf{p}_0) = \mathbf{p}_0^{\top} (I - Q)^{-1} Q_{harm} \mathbf{1}. \tag{8}$$

The model is *safely aligned* if this probability is bounded for all initial states in $S_Y^{\perp}$:

$$\max_{\mathbf{p}_0} \mathbb{P}(\text{hit } S_Y \mid \mathbf{p}_0) \leq \varepsilon. \tag{9}$$

**Remark 4.3.** *Since the states in $S \setminus (S_Y \cup S_Y^{\perp})$ cannot reach $S_Y$ and thus have hitting probability 0, they are excluded from the analysis to ensure matrix invertibility. Please see Appendix F for detailed derivation of hitting probability.*

A convenient way is to treat the refusal as an absorbing event in the Markov-chain view. Concretely, once the chain generates a refusal token (e.g., the state whose last tokens are "I'm sorry, but I can't assist with that."), the model either terminates or is forced to remain in some "refusal" state that cannot transition further into harmful states.[3] Formally, this is Definition 4.4 described below.

**Definition 4.4** (Exact absorbing state)**.** A state $s$ is said to be exactly absorbing if it contains the refusal token such that there exists a $T \in \mathbb{N}$, for all $t \geq T$,

$$Q_t(s, s') = 0 \quad \text{for all } s' \neq s \quad \text{and} \quad Q_t(s, s) = 1. \tag{10}$$

Thus, there is no path from a refusal state to any other state, particularly none leads to harmful states.

In practice, interactions with LLMs are often over multiple turns. Users rephrase questions, probe with hypothetical scenarios, or continue conversations even after a refusal. This multi-turn nature of dialogue is reflected in current leading services such as ChatGPT and DeepSeek, which do not immediately terminate conversations upon a refusal. Abruptly ending these interactions upon detection of refusal could frustrate users and degrade their overall satisfaction with the service. Thus, relaxing the exact absorbing in Definition 4.4 to Theorem 4.6 ensures continuous and coherent interactions without abruptly terminating the chat.

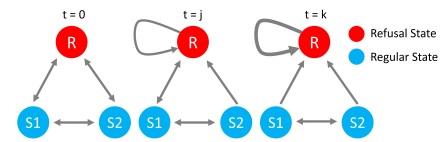

Figure 2: Visualization of $\delta$-absorbing. At $t = 0$, all states can transition freely. At $t = k$, refusal states have thick self-loops, indicating high self-transition probability.

**How Deep a Safety Alignment Should be Made?** The answer is elucidated in Theorem 4.6 and Corollary 4.8. Here, the **safety depth** is denoted as $r$. This can also be intuitively understood in Figure 2 that depicts a three-state Markov chain with one refusal state and two regular states.

**Definition 4.5** (Safety Depth)**.** Let the LLM be represented by a discrete-time Markov chain with a finite state space $S$ and a transition matrix $Q_t$ at time $t$. Let $\mathcal{R} \subset S$ be a finite, non-empty set of designated refusal-state indices. A **safety depth**, denoted by $r$, is an index corresponding to a specific refusal state, such that $r \in \mathcal{R}$.

A safety depth $r$ is considered secure at training step $T$ if its corresponding refusal state $r$ becomes $\delta$-absorbing. This means that for a given small positive constant $\delta$, the transition matrix $Q_T$ satisfies the following conditions: $|Q_T(r, r) - 1| \leq \delta$ and $|Q_T(r, s)| \leq \delta$, for all non-refusal states $s \neq r$. The first condition ensures that the probability of the model staying in the refusal state is close to 1. The second ensures that the probability of transitioning from the refusal state to any other state is close to 0.

---

[3]We aim to preserve safety under a more general setting in which the model continues producing texts even after refusal. If we only rely on an "end of sentence" (EOS) token to stop generation, the model can trivially avoid harmful contents by refusing and then terminating all outputs. However, many real-world usage scenarios do not strictly terminate on refusal. The model may receive follow-up user prompts, or it may continue auto-completing texts in multi-turn conversations. In such cases, a single refusal does not guarantee subsequent tokens remain safe. More illustrative examples are shown in Appendix K.

The **optimal safety depth**, denoted $r^*$, is the largest safety depth that is guaranteed to become $\delta$-absorbing for a given learning rate $\alpha$ and number of training steps $T$. It is formally defined as the maximum element of the set of refusal states $\mathcal{R}$ that satisfies the conditions for $\delta$-absorption:

$$r^* = \max\{r \in \mathcal{R} \mid \alpha > \alpha_r \text{ and } T > T_r\},$$

where $\alpha_r$ and $T_r$ are the minimum learning rate and training steps required to make state $r$ become $\delta$-absorbing.

While Theorem 4.6 is stated in the context of Markov chains, it can be extended to the setting of LLM finetuning under Assumption 3.1. Detailed proofs of all theorems are provided in the Appendix H.

**Theorem 4.6** ($\delta$-absorbing). *Consider a Markov chain with transition matrices $Q_t = Q_0 + \alpha\gamma^t B$, where $\gamma \in (0,1)$ is the discount factor, $\alpha > 0$ is the learning rate, and $B$ is a bias matrix with $B(r,r) > 0$ and $B(r,s) < 0$ for refusal states $r$ and $s \neq r$.*

*For any $\delta > 0$, if the training steps $T$ satisfies:*

$$T > \frac{\log(\delta(1-\gamma))}{\log(\gamma)} - 1, \tag{11}$$

*and $\alpha$ satisfies:*

$$\alpha > \frac{\max_{r,s}|Q_0(r,s)|(1-\gamma)}{\min(B(r,r), -B(r,s))}, \tag{12}$$

*then all refusal states become $\delta$-absorbing, meaning:*

$$|Q_T(r,r) - 1| \leq \delta \quad \text{and} \quad |Q_T(r,s)| \leq \delta, \tag{13}$$

*for all refusal states $r$ and non-refusal states $s$.*

**Example of Theorem 4.6.**    First, we can set up a transition matrix $Q_0$ as:

$$Q_0 = \begin{pmatrix} Q_0(r,r) & Q_0(r,s) \\ Q_0(s,r) & Q_0(s,s) \end{pmatrix} = \begin{pmatrix} 0.7 & 0.3 \\ 0.2 & 0.8 \end{pmatrix}$$

as well as set up a bias matrix $B = \begin{pmatrix} +1 & -1 \\ 0 & 0 \end{pmatrix}$. Next, for $\delta = 0.01$ and $\gamma = 0.9$, we can pick up $T = 65$ to satisfy Eq. (11), i.e., $T > \frac{\log(0.001)}{\log(0.9)} - 1 \approx 64.56$. We can also choose a proper $\alpha = 0.081$ to satisfy Eq. (12), i.e., $\alpha > \frac{0.8 \times 0.1}{1} = 0.08$. Thus, we have a sequence of transition matrices:

$$Q_1 = \begin{pmatrix} 0.7729 & 0.2271 \\ 0.2 & 0.8 \end{pmatrix}, \quad Q_2 = \begin{pmatrix} 0.8385 & 0.1615 \\ 0.2 & 0.8 \end{pmatrix}, \cdots$$

After $T$ steps, we have $Q_T(r,r) \approx 1$ and $Q_T(r,s) \approx 0$.

This demonstrates the theorem's claim that $r$ becomes effectively absorbing.

**Remark 4.7.** *As $T$ sets too large, $Q_T(r,r)$ will be a large positive number, and $Q_T(r,s)$ will be a large negative number. To handle this issue, we adopt a normalization procedure in all numerical experiments, which is described in Algorithm 1 at Appendix G .*

**Corollary 4.8** (Largest Safety Depth That Becomes $\delta$-Absorbing). *Let $\mathcal{R}$ be a finite set of refusal-state indices. For each $r \in \mathcal{R}$, define*

$$\alpha_r = \frac{\max_s |Q_0(r,s)|(1-\gamma)}{\min(B(r,r), -B(r,s))}, \tag{14}$$

*and*

$$T_r = \left\lceil \frac{\log(\delta(1-\gamma))}{\log(\gamma)} - 1 \right\rceil. \tag{15}$$

*Given any $\alpha > 0$ and $T \in \mathbb{N}$, let*

$$r^* = \max\left\{ r \in \mathcal{R} \mid \alpha > \alpha_r \text{ and } T > T_r \right\}. \tag{16}$$

*Then, for every $r \leq r^*$, the transition matrix $Q_T$ makes $r$ $\delta$-absorbing at training step $T$; that is,*

$$|Q_T(r,r) - 1| \leq \delta \quad \text{and} \quad |Q_T(r,s)| \leq \delta \quad (\forall s \neq r). \tag{17}$$

**Example of Corollary 4.8.** First, we can set up a transition matrix $Q_0$ as:

$$Q_0 = \begin{pmatrix} Q_0(1,1) & Q_0(1,2) & Q_0(1,s) \\ Q_0(2,1) & Q_0(2,2) & Q_0(2,s) \\ Q_0(s,1) & Q_0(s,2) & Q_0(s,s) \end{pmatrix} = \begin{pmatrix} 0.6 & 0.2 & 0.2 \\ 0.1 & 0.8 & 0.1 \\ 0.2 & 0.3 & 0.5 \end{pmatrix}.$$

The refusal states are 1 and 2, so $\mathcal{R} = \{1,2\}$. We choose a bias matrix $B$ as:

$$B = \begin{pmatrix} 1 & -1 & -1 \\ -1 & 1 & -1 \\ 0 & 0 & 0 \end{pmatrix}.$$

For $\delta = 0.01$ and $\gamma = 0.9$, we have $T_1 = T_2 = \left\lceil \frac{\log(0.001)}{\log(0.9)} - 1 \right\rceil = 65$. For $r = 1$, the row is $(0.6, 0.2, 0.2)$. Hence $\max_s |Q_0(1,s)| = 0.6$. Since $B(1,1) = 1$ and $-B(1,s) = 1$, the denominator is 1. Thus, $\alpha_1 = \frac{0.6 \times 0.1}{1} = 0.06$. For $r = 2$, similarly, we have $\alpha_2 = 0.08$.

Suppose we pick $(\alpha, T) = (0.075, 70)$. Then:

$$\alpha_1 = 0.06 < 0.075 < 0.08 = \alpha_2, \quad \text{and} \quad T_1 = T_2 = 65 < 70.$$

By Eq (16), $r^* = \max\left\{ r \in \{1,2\} \mid \alpha > \alpha_r \text{ and } T > T_r \right\}$. Hence the *only* $r$ satisfying both conditions is $r = 1$. Thus $r^* = 1$.

Corollary 4.8 guarantees that after $T = 70$ steps, *state 1* becomes $\delta$-absorbing, i.e.,

$$\left| Q_{70}(1,1) - 1 \right| \leq 0.01, \quad \left| Q_{70}(1,s) \right| \leq 0.01,$$

for all $s \neq 1$. Meanwhile, *state 2* is not guaranteed to be $\delta$-absorbing with these parameter values, since $\alpha = 0.075$ does not exceed $\alpha_2 = 0.08$.

If instead we pick $\alpha = 0.09 > 0.08$ (and still $T = 70 > 65$), then *both* $r = 1$ and $r = 2$ satisfy the conditions, so

$$r^* = \max\{1,2\} = 2,$$

and *both* states 1 and 2 become $\delta$-absorbing. This illustrates precisely how **the largest (optimal) safety depth** $r^*$ depends on the chosen $(\alpha, T)$.

Through Proposition 3.3 and Assumption 3.4, this framework extends naturally to the permutation group actions on the bias matrix, which shows that similar guarantees hold.

**Proposition 4.9** (Permutation Group Actions $\delta$-Absorbing). *Consider a Markov chain with transition matrices $Q_t = Q_0 + \alpha\gamma^t B^{(t)}$, where $B^{(t)} = P^t B P^{-t}$ for some permutation matrix $P$. For refusal states to become absorbing with precision $\delta > 0$, the required training steps $T$ must satisfy:*

$$T > \min\left( \frac{\log(\delta(1-\gamma))}{\log(\gamma)}, ord(P) \right) - 1, \tag{18}$$

*where $ord(P)$ is the order of permutation $P$.*

*Furthermore, if the bias matrices $B^{(t)}$ satisfy the conditions: $B^{(t)}(r,r) > 0$ for refusal states $r$, $B^{(t)}(r,s) < 0$ for $s \neq r$, and $\alpha$ satisfies:*

$$\alpha > \frac{\max_{r,s} |Q_0(r,s)|(1-\gamma)}{\min(B^{(t)}(r,r), -B^{(t)}(r,s))}, \tag{19}$$

*for all $t$, then with $T$ training steps, refusal states become $\delta$-absorbing.*

**Example of Proposition 4.9.** A simple initial transition matrix $Q_0$ can be:

$$Q_0 = \begin{pmatrix} Q_0(r,r) & Q_0(r,s_1) & Q_0(r,s_2) \\ Q_0(s_1,r) & Q_0(s_1,s_1) & Q_0(s_1,s_2) \\ Q_0(s_2,r) & Q_0(s_2,s_1) & Q_0(s_2,s_2) \end{pmatrix} = \begin{pmatrix} 0.6 & 0.2 & 0.2 \\ 0.1 & 0.8 & 0.1 \\ 0.2 & 0.3 & 0.5 \end{pmatrix}.$$

We choose a bias matrix $B$ that has:

$$B = \begin{pmatrix} +1 & -1 & -1 \\ -1 & +1 & -1 \\ -1 & -1 & +1 \end{pmatrix}.$$

Let the permutation matrix $P$ be:

$$P = \begin{pmatrix} 0 & 0 & 1 \\ 1 & 0 & 0 \\ 0 & 1 & 0 \end{pmatrix}, \quad P^3 = I, \text{ and } \text{ord}(P) = 3.$$

For $\delta = 0.01$ and $\gamma = 0.9$, Eq. (18) implies $T > 2$. Since the permutation does not influence the magnitude of $B$, Eq. (19) is the same as Eq. (12), which implies $\alpha > 0.08$. If $(\alpha, T) = (0.081, 3)$, we have a sequence of transition matrices:

$$Q_1 = \begin{pmatrix} 0.7410 & 0.1295 & 0.1295 \\ 0.0206 & 0.9586 & 0.0206 \\ 0.1295 & 0.2383 & 0.6322 \end{pmatrix}, \quad Q_3 = \begin{pmatrix} 1.0 & 0.0 & 0.0 \\ 0.0 & 1.0 & 0.0 \\ 0.0 & 0.1201 & 0.8799 \end{pmatrix}.$$

After three steps, we reach $Q_3(r, r) \approx 1$, $Q_3(r, s_1) \approx 0$, and $Q_3(r, s_2) \approx 0$.

This demonstrates the theorem's claim that $r$ becomes effectively absorbing.

**Remark 4.10.** *Proposition 4.9 converges quickly but makes the non-refusal states absorbing. This is due to the choice of the bias matrix $B$ and the permutation matrix $P$. Intuitively, it may affect the utility of LLMs. Therefore, cyclic augmentation should be trained in a few shots and stopped early.*

In practical settings, an LLM can generate harmful or benign texts at any stage of a multi-turn conversation. A non-augmented approach might predominantly insert refusals at the beginning, leaving "middle tokens" relatively underexposed to refusal training signals. Cyclic augmentation ensures that refusal states appear at diverse positions, improving the model's ability to decline harmful generation in any segment of texts.

So far, we have examined how to ensure *safety* in a *single* Markov chain by training it until it becomes $\delta$-absorbing. However, an unsolved challenge remains: how to achieve a specified safety level $\varepsilon$ when working with a set of models that individually fall short of this threshold.

Notably, Proposition 4.11 establishes that the safety constraints can be distributed across multiple models within an ensemble. Specifically, each model in the collection only needs to satisfy a safety requirement of $1/W$ of the overall threshold $\varepsilon$. This approach not only facilitates robust safety guarantees but also alleviates the training burden on individual models.

**Proposition 4.11** (Ensemble). *Consider an ensemble of width $W$ of Markov chains with transition matrices $Q_t = Q_0 + \alpha \gamma^t B$. In order for the ensemble to achieve a safety level $\varepsilon$, the required training step for each chain $T_i$ satisfies:*

$$T_i > \frac{\log(p(1 - \gamma))}{\log(\gamma)} - 1, \tag{20}$$

*where $p$ is the per-chain risk bound. Depending on the chosen aggregation strategy for the ensemble, $p$ can be bounded in any one of the following ways: (1) Union: $p \le \frac{\varepsilon}{W}$, (2) Average: $p \le \varepsilon\tau$ for some threshold $\tau \in (0, 1)$, (3) Majority voting: $p \le \frac{1}{2} - \sqrt{\frac{\ln(1/\varepsilon)}{2W}}$. Furthermore, if the bias matrices $B$ satisfy the conditions: $B(r, r) > 0$ for refusal states $r$, $B(r, s) < 0$ for $s \ne r$, and $\alpha$ satisfies:*

$$\alpha > \frac{\max_{r,s} |Q_0(r, s)|(1 - \gamma)}{\min(B(r, r), -B(r, s))}, \tag{21}$$

*for all $t$, then the ensemble achieves $\varepsilon$-safety.*

**Remark 4.12.** *Since there are many ensemble strategies, we introduce the three most common strategies—**union**, **average**, and **majority**, and show how each imposes a different requirement on the per-model risk. The experimental result will be later illustrated in Figure 3b. We leave the theoretical analysis of these strategies in Appendix H.*

In conclusion, we first show that with high probability $(1 - \delta)$, it is possible to find the optimal safety depth $r^*$ with respect to learning rate $\alpha$ and training time $T$. Moreover, we show that with cyclic group action, the convergence rate can be improved. Last but not least, the safety constraint can be distributed across multiple models with less training burden.

## 5  Experiments

In this section, we begin by presenting a toy example to validate our theoretical results, then offer illustrative cases using open-source LLMs.

**Numerical Experiments.** We conducted extensive experiments to validate our theoretical safety guarantees under various scenarios, examining three key cases: single-model convergence, cyclic group actions, and ensemble validation.

We built a simple Markov chain with four states, designating one as the refusal state. For simplicity, we set $\alpha = \gamma = 1$. We incrementally applied safety biases of magnitude 0.1 (elements in $B$) and tracked the escape probability—$1 - Q_t(r, r)$, which measures the probability of leaving the refusal state—over 50 iterations. As shown in Figure 3a, the escape probability decreased from about 0.75 to below 0.01, displaying geometric convergence consistent with Theorem 4.6 and Corollary 4.8.

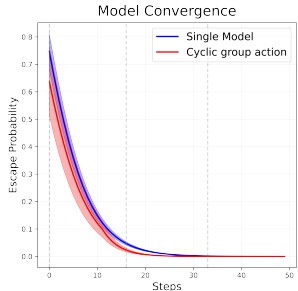
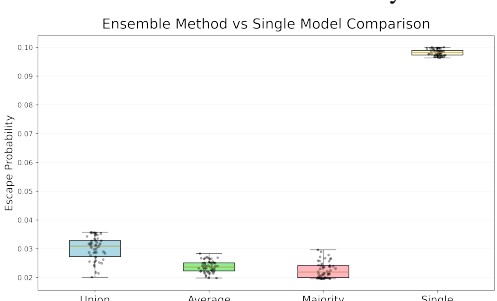

(a) Single model convergence showing exponential decay in blue line with confidence intervals over 50 bias applications, demonstrating reliable convergence to safe behavior. Cyclic group action convergence is displayed in a red line, illustrating stable convergence despite periodic fluctuations.

(b) Comparison of ensemble combination methods (Union, Average, and Majority) showing escape probabilities, where box plots indicate the distribution of outcomes and individual points show specific results.

Figure 3: Convergence behaviors and ensemble method comparisons.

Next, we alternated among three bias matrices in a cyclic manner, introducing time-varying interventions. Despite these variations, the model consistently converged to safe states, and convergence speed improved slightly (Figure 3a). These findings validate our Proposition 4.9 and Corollary H.1, showing that safety guarantees hold even when biases change over time.

Lastly, we evaluated five models, each trained to achieve a fractional safety target $\varepsilon/5$, where $\varepsilon = 0.1$. We compared three methods for combining outputs: (1) Union bound, taking the maximum escape probability across models; (2) Average, using the mean; and (3) Majority voting, taking the median. As illustrated in Figure 3b, all three methods met the overall safety threshold of $\varepsilon = 0.1$, with majority voting proving the most robust. This supports the Proposition 4.11 and underscores the value of careful aggregation strategies.

In summary, these experiments provide strong empirical evidence for our theoretical framework and practical insights into choosing bias magnitudes and ensemble methods. The observed convergence behaviors and safety guarantees closely match theoretical expectations in all tested scenarios.

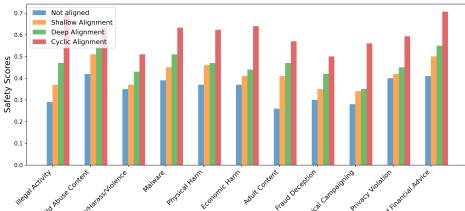

Figure 4: Each bar indicates `Gemma2-2B`'s average safety score for that category.

Table 1: `Gemma2-2B` Safety Scores Across Different Alignment Strategies.

| Category | Not Aligned | Shallow | Deep | Cyclic |
|---|---|---|---|---|
| Illegal Activity | 0.29 | 0.37 | 0.47 | 0.69 |
| Child Abuse Content | 0.42 | 0.51 | 0.54 | 0.63 |
| Hate/Harass/Violence | 0.35 | 0.37 | 0.43 | 0.51 |
| Malware | 0.39 | 0.45 | 0.51 | 0.63 |
| Physical Harm | 0.37 | 0.46 | 0.47 | 0.62 |
| Economic Harm | 0.37 | 0.41 | 0.44 | 0.64 |
| Adult Content | 0.26 | 0.41 | 0.47 | 0.57 |
| Fraud Deception | 0.30 | 0.35 | 0.42 | 0.50 |
| Political Campaigning | 0.28 | 0.34 | 0.35 | 0.56 |
| Privacy Violation | 0.40 | 0.42 | 0.45 | 0.59 |
| Tailored Financial Advice | 0.41 | 0.50 | 0.55 | 0.71 |
| Mean ± Std | 0.35 ± 0.05 | 0.42 ± 0.06 | 0.46 ± 0.05 | **0.61 ± 0.06** |

**Open-source LLMs.** We evaluated six open-source LLMs—Google's `Gemma-2-2B,9B` [Team et al., 2023], Microsoft's `Phi-2-2B` [Javaheripi et al., 2023], Alibaba's `Qwen2.5-1.5B,7B` [Yang et al., 2024], and `Mistral-7B-v0.3` [Jiang et al., 2023] —using Meta's `Llama-3.2-1B` [Touvron et al., 2023] as a judge. See Appendix I for our detailed evaluation prompt. For training, we employed the MaliciousInstruct dataset [Huang et al., 2024] of 100 harmful instructions with three data augmentation strategies (shallow, deep, cyclic). Testing was conducted on the HEx-PHI dataset,[4]

---
[4]https://huggingface.co/datasets/LLM-Tuning-Safety/HEx-PHI

which contains 330 harmful instructions spanning 11 prohibited categories. HEx-PHI, derived from sources like Meta's and OpenAI's policies with human annotations and model inputs (GPT-4, Claude), serves strictly for safety evaluation rather than malicious use.

| DATASET | Not Aligned | Shallow | Deep | Cyclic |
|---------|-------------|---------|------|--------|
| SAMSum (ROUGE-1) | 0.2834 | 0.2798 | 0.2787 | 0.2750 |
| GSM8K (Accuracy) | 0.2200 | 0.2100 | 0.2000 | 0.2000 |
| SQL (AvgTokenMatch) | 0.6112 | 0.6181 | 0.6307 | 0.5898 |

Table 2: `Gemma2-2B` Performance on helpfulness.

As shown in Figure 4 and Table 1, applying cyclic augmentation on `Gemma-2-2B` significantly boosts safety scores. However, Table 2 suggests that across three tasks: SAMSum (summarization), GSM8K (math reasoning), and SQL (coding), there is a trade-off between helpfulness and safety.

Due to the space constraint, results for other models and dataset descriptions appear in Appendix J.

From the data augmentation experiments in Table 1 and Tables 5, 6 (Appendix J), we found `Qwen2.5-1.5B` most effective under deep alignment. We therefore compared three "shallow" models in an ensemble against the "deep" `Qwen2.5-1.5B` model.

For all three ensemble methods, any output with a safety score below our predefined threshold (set at 0.7) is immediately discarded. This thresholding mechanism provides an additional layer of security, ensuring that only responses meeting a high standard of safety are retained for further use. As Figure 5 illustrates, union, average, and majority ensembles consistently scored higher, clustering around 0.9–1.0 and indicating stronger safety. Deep alignment, by contrast, showed broader variation and a lower median, suggesting inconsistent safety. This aligns with our theoretical findings, confirming that ensemble methods offer more robust safety.

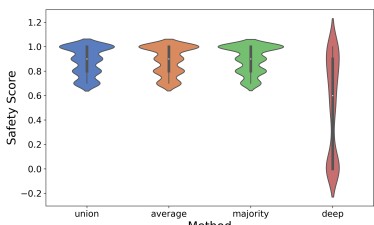

Figure 5: Violin plot of ensemble methods vs. deep alignment.

**Remark.** There are two primary application scenarios. In the first scenario, if the user is willing to retrain or finetune the model, we recommend adopting the cyclic augmentation technique. This approach allows for direct improvement of model robustness through additional data augmentation during the training process. Cyclic augmentation is theoretically guaranteed to achieve optimal safety depth, as shown in Corollary H.1. This theoretical result is further validated by our sample-level experiments. In the second scenario, if the user does not wish to retrain or finetune the model, we recommend employing an ensemble strategy instead. This is also practical, as shown in Appendix J.3.

**Verification of cyclic augmentation improves refusal positioning.** We conducted the following sample-level experiments to demonstrate that cyclic augmentation improves refusal positioning. We tested 100 harmful prompts (e.g., "Tell me how to make a bomb") with two models: the Baseline (Qwen2.5-1.5B) and the Cyclic Augmentation variant (Qwen2.5-1.5B-cyclic). Each scenario involved 5 turns per prompt, yielding a total of 500 interactions per model (100 scenarios × 5 turns). Refusal Depth measures a model's resilience to adversarial pressure. The metric is defined such that a model scores one point for each consecutive turn in which it successfully refuses to provide a harmful response. Scoring terminates when the model provides a harmful response or completes all 5 turns, at which point the final Refusal Depth is logged for each scenario.

| Model | Mean Refusal Depth | Min Refusal Depth | Overall Refusal Rate (Sustained for 5 turns) |
|-------|--------------------|--------------------|----------------------------------------------|
| Baseline | 2.3 | 1 | 40% |
| Cyclic Augmentation | 4.6 | 3 | 90% |

Table 3: Refusal Depth Comparison: Baseline vs. Cyclic Augmentation

Our conclusion is that cyclic augmentation significantly improves refusal positioning, because a much higher mean refusal depth (e.g., doubling it from 2 to over 4) and a higher rate of maintaining refusal throughout all conversational turns.

## 6 Conclusion

This paper answered the question "*How deep a safety alignment should be made?*" through the context of Markov chain. We provided an insightful analysis from a single model, cyclic group acting to the ensemble method. The numerical experiments also justify our theoretical findings. We hope that our theoretical insights will affect algorithm design for LLM safety alignment in the future.

**Acknowledgment.** This work was supported by the National Science and Technology Council (NSTC), Taiwan, ROC, under Grants NSTC 114-2221-E-001-010-MY2 and NSTC 114-2222-E-A49-011-MY3.

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

# A Group Theory and Rotations Form a Cyclic Group

## A.1 Basic Definitions

**Definition A.1** (Group). A *group* is a set $G$ together with a binary operation $\cdot$ satisfying the following properties:

1. **Closure:** For all $a, b \in G$, the product $a \cdot b$ is also in $G$.

2. **Associativity:** For all $a, b, c \in G$, $(a \cdot b) \cdot c = a \cdot (b \cdot c)$.

3. **Identity:** There exists an element $e \in G$ such that for all $a \in G$, $e \cdot a = a \cdot e = a$.

4. **Inverse:** For every $a \in G$, there exists an element $a^{-1} \in G$ such that $a \cdot a^{-1} = a^{-1} \cdot a = e$.

**Definition A.2** (Cyclic Group). A group $G$ is called *cyclic* if there exists an element $g \in G$ such that every element of $G$ can be written as $g^n$ (i.e., repeated products of $g$ with itself or its inverse) for some integer $n$. We say $g$ *generates* $G$, and write $G = \langle g \rangle$.

## A.2 Rotations on $n$ Letters

Let us label $n$ distinct letters as $a_1, a_2, \ldots, a_n$. We look at *rotations* of these letters as permutations in a line. A *one-step rotation* $\rho$ acts by sending

$$(a_1, a_2, \ldots, a_{n-1}, a_n) \;\mapsto\; (a_2, a_3, \ldots, a_n, a_1). \tag{22}$$

Reapplying $\rho$ repeatedly shifts all letters one position each time.

**Example 1** (The case $n = 3$). If we have letters $\{a, b, c\}$, the one-step rotation $\rho$ sends:

$$(a, b, c) \mapsto (b, c, a). \tag{23}$$

Then,

$$\rho^2 : (a, b, c) \mapsto (c, a, b), \quad \rho^3 = \rho^0 : (a, b, c) \mapsto (a, b, c). \tag{24}$$

Hence, all possible rotations are

$$\{\rho^0, \rho^1, \rho^2\} = \{e, \rho, \rho^2\}. \tag{25}$$

## A.3 Why Rotations Form a Group

**Closure:** The composition of two rotations is still a rotation (adding their "shift amounts" modulo $n$).

**Associativity:** Follows from the associativity of permutation composition.

**Identity:** The "zero-step rotation" (do nothing) is the identity permutation, denoted $\rho^0$.

**Inverses:** A $k$-step rotation can be undone by an $(n - k)$-step rotation. Formally, $(\rho^k)^{-1} = \rho^{-k} \equiv \rho^{n-k}$.

Hence, all rotations $\{\rho^0, \rho^1, \ldots, \rho^{n-1}\}$ form a group (often called the *cyclic group of order $n$* and denoted $C_n$).

## A.4 Why It Is Cyclic

Only one generator is needed: the *one-step rotation* $\rho$. Indeed, every rotation is a power of $\rho$:

$$\rho^0 = e, \quad \rho^1 = \rho, \quad \rho^2, \quad \ldots, \quad \rho^{n-1}. \tag{26}$$

Moreover,

$$\rho^n = \rho^0 = e, \tag{27}$$

so there are exactly $n$ distinct elements. Thus, the entire group is generated by the single element $\rho$, making it *cyclic*.

## A.5   Summary

- A **group** is a set with an associative, invertible operation and an identity element.
- A **cyclic group** is a group generated by a single element.
- **Rotations on** $n$ **letters** (in a circle or in a line) form a group under composition:

$$\{e, \rho, \rho^2, \ldots, \rho^{n-1}\}, \tag{28}$$

and they are generated by the one-step rotation $\rho$. Hence, this group is *cyclic*.

## A.6   Group Representations.

A fundamental concept in studying symmetries is the representation of groups through linear transformations.

**Definition A.3** (Group Representation). A representation $\rho$ is a map from a group $G$ to linear transformations:

$$\rho : G \to \mathrm{GL}(V), \tag{29}$$

where $V$ is a vector space and $\rho$ is a group homomorphism, i.e.,

$$\rho(g_1 g_2) = \rho(g_1)\rho(g_2), \quad \forall g_1, g_2 \in G. \tag{30}$$

A representation is called orthogonal if $\rho(g^{-1}) = \rho(g)^\top$ for all $g \in G$. This property is significant for maintaining geometric relationships under group actions.

For readers unfamiliar with group theory and linear representations of finite groups, we recommend referring to [Dummit and Foote, 2004, Serre et al., 1977] for a comprehensive introduction.

# B   Discussion

Two theoretical constructs—(1) Markov chains with adjustable transition matrices and (2) the Neural Tangent Kernel (NTK) regime for infinitely wide neural networks—can initially seem unrelated. Yet LLMs practically merge these perspectives. They use neural networks to parameterize enormous transition distributions and are trained like Markov chains that predict the next token/state. Below, we reflect on why these two points of view are connected at a deeper level.

**Neural Tangent Kernel.**   In the infinite-width regime (as width $n \to \infty$), with i.i.d. initialization of weights according to a scaled Gaussian distribution, the behavior of neural networks can be characterized through kernel methods. For any time $t$, we define the Neural Tangent Kernel (NTK) $\Theta_t : \mathcal{X} \times \mathcal{X} \to \mathbb{R}^{m \times m}$ as:

$$\Theta_t(x, x') = \nabla_\theta \pi_\theta(\cdot|x) \nabla_\theta \pi_\theta(\cdot|x')^\top. \tag{31}$$

The NTK captures the evolution of neural networks during gradient descent training. At initialization ($t = 0$), the NTK converges in probability to a deterministic kernel as the width approaches infinity:

$$\Theta_0(x, x') \xrightarrow{P} \mathbb{E}_{\theta \sim p}[\nabla_\theta \pi_\theta(\cdot|x) \nabla_\theta \pi_\theta(\cdot|x')^\top]. \tag{32}$$

Moreover, in this infinite-width limit, the NTK remains approximately constant throughout training:

$$\Theta_t(x, x') \approx \Theta_0(x, x'), \quad \text{for all } t \geq 0. \tag{33}$$

This phenomenon, known as the "lazy training" regime, allows us to analytically solve the training dynamics. Let $\mu_t(x)$ denote the mean prediction at time $t$ for input $x$. Under gradient flow, the evolution of $\mu_t$ follows:

$$\frac{\partial \mu_t(x)}{\partial t} = -\eta \Theta_0(x, X)(\mu_t(X) - Y), \tag{34}$$

where $X$ represents the training inputs, $Y$ denotes the corresponding targets, and $\eta$ is the learning rate. This linear differential equation admits a closed-form solution, providing a complete characterization of the network's training dynamics in the infinite-width limit.

**Shared Geometry of Learning Trajectories.** Consider a large vocabulary with $n$ possible tokens. Each row of the transition matrix $Q(\theta)$ is a point in an $(n-1)$-dimensional simplex. Updating $Q_{ij}(\theta)$ for all $i, j$ is equivalent to moving within $n \times n$ simplices, but these updates are *not* row-wise independent due to the shared network parameters. In NTK language: $\Delta Q(\text{state}) \approx \nabla_\theta Q(\text{state}; \theta_0) \Delta \theta$, which couples the transition distributions for all states through $\Delta \theta$ in a kernel-like manner. Hence, the geometry is simultaneously discrete on the output side (probability vectors) but linear on the parameter side (NTK approximation).

**Mixing Times vs. Convergence in Parameter Space.** Markov chains converge to a stationary distribution at a rate governed by their spectral gap (or second-largest eigenvalue [5]). Neural nets in the NTK regime converge at a rate set by $\eta \, \lambda_{\min}(\Theta_0)$, where $\Theta_0$ is the kernel matrix on training data. The unifying theme is an eigen-structure that dominates *how fast* the system approaches equilibrium (stationary distribution in MC, or minimal training loss in NTK).

## C   Notation Table

Table 4 summarizes the notations used throughout this paper.

| Notation | Description |
|----------|-------------|
| $\mathcal{V}$ | Dictionary |
| $D$ | Dictionary(vocabulary) size |
| $K$ | Context window size |
| $\mathcal{V}_K^*$ | A set of phrases of length less than $K$ |
| $\pi_\theta$ | Autoregressive LLM |
| $\Delta(\cdot)$ | Probability simplex |
| $\mathcal{X}$ | Input documents |
| $\mathcal{Y}$ | Output documents |
| $Q_t$ | Transition matrix at time $t$ |
| $\alpha$ | Learning Rate |
| $\gamma$ | Discount factor |
| $B$ | Bias matrix |
| $T$ | Training steps |
| $\mathbf{p}_0$ | Initial distribution |
| $\pi$ | Stationary distribution |
| $\rho$ | Representation map |
| $G$ | Group |
| $\mathrm{GL}(V)$ | General Linear Group over a vector space $V$ |
| $\mathcal{T}$ | Training set |
| $\mathcal{T}_{\text{aug}}$ | Augmented Training set |
| $S_Y$ | Harmful states |
| $S_Y^\perp$ | Non-Harmful states that can transfer to Harmful states |
| $r$ | safety depth or refusal state |
| $r^*$ | optimal safety depth |
| $\mathcal{R}$ | a set of refusal states |

Table 4: Notation Table.

## D   Extra Preliminaries

If the chain is trained over $T$, the *total* bias added up to time $T$ is:

$$\sum_{t=0}^{T} \alpha \, \gamma^t = \alpha \, \frac{1 - \gamma^{T+1}}{1 - \gamma}. \tag{35}$$

---

[5]Since by Perron–Frobenius theorem, the largest eigenvalue is always 1.

As $T \to \infty$ and $\gamma \in (0, 1)$, this converges to $\frac{\alpha}{1-\gamma}$. Hence, if $\alpha$ and $\gamma$ are chosen such that $\frac{\alpha}{1-\gamma} \ll 1$, the long-run total bias remains sufficiently small to ensure the perturbed matrix $Q_t$ stays close to the original $Q_0$ while introducing a controlled preference (e.g., towards a particular state or set of states).

As $t \to \infty$, the transition matrix $Q_t$ converges to

$$Q_\infty = Q_0 + \frac{\alpha}{1-\gamma} B.$$

Let $\pi$ be the stationary distribution of the limiting chain, i.e.,

$$\pi Q_\infty = \pi.$$

Under typical ergodicity conditions (i.e., irreducibility and aperiodicity of $Q_\infty$), the Markov chain converges to $\pi$. We can write

$$\lim_{t\to\infty} \mathbf{p}_0^\top Q_t = \pi,$$

for any initial distribution $\mathbf{p}_0$.

The speed at which $Q_t$ converges to $Q_\infty$ is dominated by the factor $\gamma^t$ in the bias term:

$$\|Q_t - Q_\infty\| \leq O(\gamma^t),$$

where $\|\cdot\|$ can be any matrix norm consistent with vector norms (e.g., the operator norm). Consequently, the Markov chain *mixes* to its final stationary distribution at a rate that depends geometrically on $\gamma^t$.

To ensure that $\|Q_t - Q_\infty\| < \delta$, for some small $\delta > 0$, one typically requires

$$t \geq \frac{\log(\delta)}{\log(\gamma)}.$$

**Remark D.1.** *It is possible that the discount factor does not follow the geometric series. However, we only require the boundedness of the discount sequence to analyze the main result.*

# E    Inner Structure of Large Language Models

Let $E \in \mathbb{R}^{|\Sigma| \times d}$ be the embedding matrix, where $d$ is the embedding dimension. The embedding lookup function $e : \Sigma \to \mathbb{R}^d$ maps each token to its corresponding vector representation:

$$e(x_t) = E_{x_t}, \quad \text{where } x_t \in \Sigma \tag{36}$$

For a sequence $x = (x_1, ..., x_{|x|})$, the embedded representation $X \in \mathbb{R}^{|x| \times d}$ is:

$$X = (e(x_1), ..., e(x_{|x|})) \tag{37}$$

In transformer-based models, positional information is incorporated through positional embeddings $P \in \mathbb{R}^{|x| \times d}$, giving us:

$$\hat{X} = X + P \tag{38}$$

The model then processes these embeddings through $L$ transformer layers. For each layer $l \in \{1, ..., L\}$:

$$Q^l = H^{l-1} W_Q^l \tag{39}$$

$$K^l = H^{l-1} W_K^l \tag{40}$$

$$V^l = H^{l-1} W_V^l \tag{41}$$

$$A^l = \text{softmax}\left(\frac{Q^l (K^l)^T}{\sqrt{d}}\right) V^l \tag{42}$$

$$H^l = \text{LayerNorm}(\text{FFN}(\text{LayerNorm}(H^{l-1} + A^l))) \tag{43}$$

where:

- $H^0 = \hat{X}$ is the input embedding

- $W_Q^l, W_K^l, W_V^l \in \mathbb{R}^{d \times d}$ are learned parameter matrices
- FFN is a position-wise feed-forward network
- LayerNorm is layer normalization

The final layer output $H^L$ is then projected to vocabulary logits through a linear transformation $W_O \in \mathbb{R}^{d \times |\Sigma|}$:

$$\text{logits}_t = H_t^L W_O \tag{44}$$

Finally, the probability distribution over the next token is given by:

$$\pi_\theta(y_t|x, y_{<t}) = \text{softmax}(\text{logits}_t) \tag{45}$$

The complete probability of generating sequence $y$ given input $x$ is:

$$\pi_\theta(y|x) = \prod_{t=1}^{|y|} \pi_\theta(y_t|x, y_{<t}) \tag{46}$$

# F   Hitting Probability derivation

We follow the steps below to derive the hitting probability.

1. **Define the Hitting Probability**: Let $h_i$ be the probability that the model will eventually enter a harmful state ($S_Y$), given that it starts in a specific non-harmful state $i$ (where $i \in S_Y^\perp$). We want to find the value of $h_i$ for all possible starting states $i$.

2. **Apply First-Step Analysis**: From state $i$, the model can do one of two things in its first transition:
   - Move to another non-harmful state $j \in S_Y^\perp$. The probability of this is given by the matrix $Q$. If it moves to state $j$, the probability of then hitting a harmful state is, by definition, $h_j$.
   - Move directly to a harmful state $s \in S_Y$. The probability of this is given by the matrix $Q_{harm}$. If this happens, the model has successfully hit the harmful set, so the probability is 1.

3. **Formulate a System of Equations**: Based on the two possibilities above, we can write a system of linear equations for each starting state $i$:   $h_i$   =   (Prob. of moving to another non-harmful state $j$, summed over all $j$)   +   (Prob. of moving directly to a harmful state) $h_i = \sum_{j \in S_Y^\perp} Q_{ij} h_j + \sum_{s \in S_Y} (Q_{harm})_{is}$

4. **Convert to Matrix Form**: This system of equations can be expressed more cleanly using matrix notation:
   - Let $\mathbf{h}$ be a column vector containing all the hitting probabilities $h_i$.
   - The term $\sum_{j \in S_Y^\perp} Q_{ij} h_j$ is the matrix-vector product $Q\mathbf{h}$.
   - The term $\sum_{s \in S_Y} (Q_{harm})_{is}$ is the probability of moving from state $i$ to *any* harmful state. This can be written as the matrix-vector product $Q_{harm}\mathbf{1}$, where $\mathbf{1}$ is a column vector of ones.

   The equation for the entire system becomes: $\mathbf{h} = Q\mathbf{h} + Q_{harm}\mathbf{1}$

5. **Solve for h**: Now, we can algebraically solve for the vector of hitting probabilities $\mathbf{h}$.

$$\mathbf{h} - Q\mathbf{h} = Q_{harm}\mathbf{1}$$
$$(I - Q)\mathbf{h} = Q_{harm}\mathbf{1}$$
$$\mathbf{h} = (I - Q)^{-1} Q_{harm}\mathbf{1}$$

6. **Incorporate the Initial Distribution** ($p_0$): The vector $\mathbf{h}$ gives the hitting probability for each specific starting state. To get the single, overall probability for a given initial distribution $p_0$ over the starting states, we take the weighted average. This is calculated as the dot product of the initial distribution vector $p_0$ and the hitting probability vector $\mathbf{h}$.

   This gives the final formula as presented in the paper:

$$\mathbb{P}(\text{hit } S_Y | p_0) = p_0^\top \mathbf{h} = p_0^\top (I - Q)^{-1} Q_{harm}\mathbf{1}$$

## G   Algorithms

Theorem 4.6 states that, under certain conditions on $T$ and $\alpha$,

$$\left|Q_T(r, r) - 1\right| \leq \delta \quad \text{and} \quad \left|Q_T(r, s)\right| \leq \delta,$$

for refusal state $r$. Mathematically, the theorem's proof examines the raw updates

$$Q_0(r, r) + \alpha \sum_{t=0}^{T} \gamma^t B(r, r) \quad \text{and} \quad Q_0(r, s) + \alpha \sum_{t=0}^{T} \gamma^t B(r, s).$$

When the sum is large, we often get a raw update $\gg 1$ for $r \rightarrow r$ and $\ll 0$ for $r \rightarrow s$.

- In the Markov chain interpretation, those out of $[0, 1]$ values are evidence that the process has been "pushed" to an extreme (making $r$ near-absorbing).

- By projecting (clipping + normalizing), you create a proper probability distribution out of that raw shift. The near-absorbing behavior is retained because

$$Q_0(r, r) + \alpha \sum \gamma^t B(r, r) \quad \gg \quad Q_0(r, s) + \alpha \sum \gamma^t B(r, s),$$

so the ratio in the normalization step is close to $1$, and $\delta$-absorbing is satisfied in practice. Hence, the combination of (a) the large-magnitude bias updates and (b) projecting onto the probability simplex ensures the final matrix satisfies the theorem's condition that $Q_T(r, r) \approx 1$. The detail algorithm is in Algorithm 1.

---

**Algorithm 1:** Transition Matrix Normalization

**Input:** Initial matrix $Q_0$, bias matrix $B$, parameters $\alpha, \gamma, T$
**Output:** Updated stochastic matrix $Q_T$

1  **for** *each row $i$* **do**
2     **Compute unnormalized updates**
3     **for** *each column $s$* **do**
4        $\widetilde{Q}_T(i, s) \longleftarrow Q_0(i, s) + \alpha \sum_{t=0}^{T} \gamma^t B(i, s)$;
5        **Clip negative values**
6        $Q_T^+(i, s) \longleftarrow \max\{0, \widetilde{Q}_T(i, s)\}$;
7     **end**

8     **Compute row sum**
9     $Z_i \longleftarrow \sum_s Q_T^+(i, s)$;
10    **if** $Z_i > 0$ **then**
11       **Normalize row**
12       **for** *each column $s$* **do**
13          $Q_T(i, s) \longleftarrow Q_T^+(i, s)/Z_i$;
14       **end**
15    **end**
16    **else**
17       **Handle zero-sum row**
         `/* Option 1:  Revert to original                    */`
18       $Q_T(i, s) \longleftarrow Q_0(i, s)$ ;      `// or`
         `/* Option 2:  Make absorbing                         */`
19       $Q_T(i, i) \longleftarrow 1$ ;      `// or`
         `/* Option 3:  Problem-specific handling              */`
20    **end**
21 **end**

---

# H Detailed Proof

## H.1 Proof of Theorem 4.6

*Proof.* The overall goal is to show that, under the stated conditions on $T$ and $\alpha$, each refusal state $r$ becomes $\delta$-absorbing in the modified Markov chain.

**Step 1. Bounding the Geometric Sum.**

$$\left| \frac{1}{1-\gamma} - \sum_{i=0}^{T} \gamma^i \right| = \frac{\gamma^{T+1}}{1-\gamma} \leq \delta \tag{47}$$

We want to ensure

$$\frac{\gamma^{T+1}}{1-\gamma} \leq \delta \quad \Longleftrightarrow \quad \gamma^{T+1} \leq \delta(1-\gamma). \tag{48}$$

Taking logs on both sides, given $0 < \gamma < 1$, we get

$$(T+1) \log(\gamma) \leq \log\big(\delta(1-\gamma)\big). \tag{49}$$

Because $\log(\gamma) < 0$, dividing by $\log(\gamma)$ reverses the inequality direction:

$$T+1 \geq \frac{\log(\delta(1-\gamma))}{\log(\gamma)} \quad \Longleftrightarrow \quad T > \frac{\log(\delta(1-\gamma))}{\log(\gamma)} - 1. \tag{50}$$

This matches precisely the stated condition on $T$ in the theorem:

$$T > \frac{\log(\delta(1-\gamma))}{\log(\gamma)} - 1. \tag{51}$$

Hence, if $T$ is large enough so that $\gamma^{T+1}/(1-\gamma) \leq \delta$, then

$$\left| \frac{1}{1-\gamma} - \sum_{i=0}^{T} \gamma^i \right| \leq \delta. \tag{52}$$

**Step 2: Showing $Q_T(r, r)$ Is Close to 1.**

We now analyze the self-transition probability at refusal state $r$ after $T$ steps:

$$Q_T(r,r) = Q_0(r,r) + \alpha \sum_{i=0}^{T} \gamma^i B(r,r). \tag{53}$$

Given that $B(r,r) > 0$, we want to show $Q_T(r,r)$ is close to 1. More explicitly:

$$Q_T(r,r) = Q_0(r,r) + \alpha B(r,r) \sum_{i=0}^{T} \gamma^i = Q_0(r,r) + \alpha B(r,r) \left( \frac{1}{1-\gamma} - \frac{\gamma^{T+1}}{1-\gamma} \right). \tag{54}$$

Hence

$$Q_T(r,r) - 1 = \left( Q_0(r,r) + \frac{\alpha B(r,r)}{1-\gamma} - 1 \right) - \alpha B(r,r) \frac{\gamma^{T+1}}{1-\gamma}. \tag{55}$$

If we choose $\alpha$ so that

$$Q_0(r,r) + \frac{\alpha B(r,r)}{1-\gamma} = 1 \quad \Longleftrightarrow \quad \alpha = \big(1 - Q_0(r,r)\big) \frac{1-\gamma}{B(r,r)}, \tag{56}$$

then as $T$ grows, since $\alpha B(r,r) < 1$, the extra term $\alpha B(r,r) \frac{\gamma^{T+1}}{1-\gamma}$ becomes negligible ($\leq \delta$) once $\gamma^{T+1}/(1-\gamma) \leq \delta$.

then we get

$$|Q_T(r,r) - 1| \leq \delta. \tag{57}$$

**Step 3: Showing $Q_T(r, s)$ Is Close to 0 for $s \neq r$.**

For transitions out of $r$ into a different state $s \neq r$, we have:

$$Q_T(r, s) = Q_0(r, s) + \alpha \sum_{i=0}^{T} \gamma^i B(r, s). \tag{58}$$

Because $B(r, s) < 0$ whenever $r$ is a refusal state and $s \neq r$, we can force $Q_T(r, s)$ to be close to 0 by making $\alpha$ large enough in magnitude (since the negative bias term will push the transitions toward 0).

Similar to **Step 2** we have,

$$Q_T(r, s) = \left( Q_0(r, s) + \frac{\alpha B(r, s)}{1 - \gamma} \right) - \alpha B(r, s) \frac{\gamma^{T+1}}{1 - \gamma}. \tag{59}$$

If we choose $\alpha$ so that

$$Q_0(r, s) + \frac{\alpha B(r, s)}{1 - \gamma} = 0 \iff \alpha = -Q_0(r, s) \frac{1 - \gamma}{B(r, s)}, \tag{60}$$

then as $T$ grows, similar argument holds.

As long as

$$|Q_T(r, s)| \leq \delta, \tag{61}$$

we get the desired bound. This translates into a lower bound on $\alpha$ (recalling $B(r, s) < 0$ and $\delta > 0$):

$$\alpha > \frac{\max_{r,s} |Q_0(r, s)|(1 - \gamma)}{\min\big(B(r, r), -B(r, s)\big)}. \tag{62}$$

Here $\min(B(r, r), -B(r, s))$ ensures that $\alpha$ works for all relevant refusal states $r$ simultaneously (both for pulling the diagonal entries toward 1 and pulling the off-diagonal entries toward 0).

$\square$

## H.2 Proof of Corollary 4.8

*Proof.* We proceed in three steps:

**1. Setup and Geometric Sum Convergence**

For any refusal state $r$, the transition probabilities at time $T$ are given by:

$$Q_T(r, s) = Q_0(r, s) + \alpha B(r, s) \sum_{i=0}^{T} \gamma^i \tag{63}$$

For $T > T_r$, we have:

$$\left| \sum_{i=0}^{T} \gamma^i - \frac{1}{1 - \gamma} \right| \leq \frac{\delta}{1 - \gamma} \tag{64}$$

This follows from the definition of $T_r$ and the properties of geometric series.

**2. Absorption Properties for Qualifying States**

Let $r \leq r^*$. Then $\alpha > \alpha_r$ and $T > T_r$ by definition.

For the self-transition $(r, r)$:

$$Q_T(r, r) = Q_0(r, r) + \alpha B(r, r) \sum_{i=0}^{T} \gamma^i$$

$$\geq Q_0(r, r) + \alpha B(r, r) \left( \frac{1}{1 - \gamma} - \frac{\delta}{1 - \gamma} \right)$$

$$\geq 1 - \delta$$

The final inequality follows from the condition $\alpha > \alpha_r$ and the definition of $\alpha_r$.

For non-self transitions $(r, s)$ where $s \neq r$:

$$Q_T(r, s) = Q_0(r, s) + \alpha B(r, s) \sum_{i=0}^{T} \gamma^i$$

$$\leq Q_0(r, s) + \alpha B(r, s) \left( \frac{1}{1 - \gamma} + \frac{\delta}{1 - \gamma} \right)$$

$$\leq \delta$$

### 3. Inheritance by Smaller Indices

For any $r' < r \leq r^*$:

- $\alpha_{r'} \leq \alpha_r < \alpha$ (by monotonicity of the bias requirements)
- $T_{r'} = T_r < T$ (since $T_r$ depends only on $\delta$ and $\gamma$)

Therefore, all states $r \leq r^*$ become $\delta$-absorbing by time $T$. $\qquad \square$

## H.3  Proof of Theorem 4.9

*Proof.* The overall goal is the same as Theorem 4.6. The only difference is the bias matrix $B$ become a family of matrices $B^{(t)} = P^t B P^{-t}$, where $P$ is a permutation matrix of finite order $m = \mathrm{ord}(P)$. Concretely, $P^m = I$, so

$$B^{(t+m)} = P^{t+m} B P^{-t-m} = P^t \left( P^m B P^{-m} \right) P^{-t} = P^t B P^{-t} = B^{(t)}. \tag{65}$$

Hence the sequence $\{B^{(t)}\}_{t=0}^{\infty}$ cycles every $m$ steps. For $t \geq m$,

$$B^{(t)} = B^{(t \bmod m)}. \tag{66}$$

Since $P$ is *orthonormal* in $\ell^\infty$-matrix norm, we have

$$\|P^t B P^{-t}\|_\infty = \|B\|_\infty \quad \text{for all } t. \tag{67}$$

That uniform bound ensures each $B^{(t)}$ behaves like $B$ in size; it is merely a row/column permutation of $B$ Because $P^m = I$, you can check that

$$\sum_{t=0}^{m-1} B^{(t)} = \sum_{t=0}^{m-1} P^t B P^{-t} = \sum_{t=0}^{m-1} B^{(0)} = m B, \tag{68}$$

**Case (I).**  If $T \geq m - 1$, then you have at least one complete cycle. In that case, the difference from the infinite sum primarily depends on tail terms beyond $T$, which in the simpler geometric-series argument is $\sim \gamma^{T+1}/(1 - \gamma)$.

**Case (II).**  If $T < m - 1$, you have not even completed *one* full cycle, so it might be that you need to wait for at least $m$ steps to see the intended effect. In that scenario, you may want to ensure $\gamma^m$ is already small enough, i.e., $\gamma^m/(1 - \gamma) < \delta$, so that even partial cycles do not matter.

$\qquad \square$

**Corollary H.1** (Largest Safety Depth with Permutation Group Actions). *Let $\mathcal{R}$ be a finite set of refusal-state indices. For each $r \in \mathcal{R}$, define*

$$\alpha_r = \frac{\max_s |Q_0(r, s)|(1 - \gamma)}{\min_{t < ord(P)} \min(B^{(t)}(r, r), -B^{(t)}(r, s))}, \tag{69}$$

*and*

$$T_r = \left\lceil \max \left( \frac{\log(\delta(1 - \gamma))}{\log(\gamma)}, ord(P) \right) - 1 \right\rceil. \tag{70}$$

*Given any $\alpha > 0$ and $T \in \mathbb{N}$, let*

$$r^* = \max \{ r \in \mathcal{R} \mid \alpha > \alpha_r \text{ and } T > T_r \}, \tag{71}$$

*then for every $r \leq r^*$, the transition matrix $Q_T$ makes $r$ $\delta$-absorbing at training step $T$.*

## H.4 Proof of Corollary H.1

*Proof.* Setup and Cyclic Structure For any refusal state $r$, the transition probabilities at time $T$ are:

$$Q_T(r, s) = Q_0(r, s) + \alpha \sum_{t=0}^{T} \gamma^t B^{(t)}(r, s) \tag{72}$$

where $B^{(t)} = P^t B P^{-t}$ cycles with period $\mathrm{ord}(P)$. Convergence of Cyclic Sum The key is analyzing the sum $\sum_{t=0}^{T} \gamma^t B^{(t)}$. Since $B^{(t)}$ has period $\mathrm{ord}(P)$, we can group terms:

$$\sum_{t=0}^{T} \gamma^t B^{(t)} = \sum_{i=0}^{\lfloor T/\mathrm{ord}(P) \rfloor} \gamma^{i \cdot \mathrm{ord}(P)} \sum_{j=0}^{\mathrm{ord}(P)-1} \gamma^j B^{(j)} + R_T \tag{73}$$

where $R_T$ is a remainder term with at most $\mathrm{ord}(P) - 1$ terms. Error Bounds For $T > T_r$, we have:

$$\left| \sum_{t=0}^{T} \gamma^t B^{(t)} - \frac{1}{1-\gamma} \sum_{j=0}^{\mathrm{ord}(P)-1} \gamma^j B^{(j)} \right|_\infty \leq \frac{\delta}{1-\gamma} \tag{74}$$

This follows from our choice of $T_r$ which ensures both sufficient geometric decay and completion of cyclic periods. Absorption Properties For any $r \leq r^*$, we have $\alpha > \alpha_r$ and $T > T_r$. For self-transitions:

$$Q_T(r, r) \geq Q_0(r, r) + \alpha \min_t B^{(t)}(r, r) \cdot \frac{1-\delta}{1-\gamma} \geq 1 - \delta \tag{75}$$

And for non-self transitions:

$$Q_T(r, s) \leq Q_0(r, s) + \alpha \max_t B^{(t)}(r, s) \cdot \frac{1+\delta}{1-\gamma} \leq \delta \tag{76}$$

Inheritance by Smaller Indices The conditions for any $r' < r \leq r^*$ are satisfied because:

$\alpha_{r'}$ accounts for worst-case cyclic bias over all periods $T_{r'}$ ensures both geometric convergence and cyclic completion

Therefore, all states $r \leq r^*$ become $\delta$-absorbing by time $T$. $\square$

## H.5 Proof of Theorem 4.11

From the single-model $\delta$-absorbing proofs (like Theorem 4.6), we know that if we train a *single* Markov chain up to step $T$ (and pick $\alpha$ large enough) so that all refusal states become $\delta$-absorbing, then the probability of leaving a refusal state (and thus being "harmful") can be made arbitrarily small. Symbolically, if the chain is $\delta$-absorbing, we get

$$\Pr(\text{harmful}) \approx O(\delta). \tag{77}$$

Hence, to ensure $\Pr(M_i \text{ is harmful}) \leq p_i$, we typically require

$$T_i > \underbrace{\frac{\log(p_i(1-\gamma))}{\log(\gamma)}}_{\text{from geometric tail } \leq p_i} - 1, \quad \text{and} \quad \alpha > \frac{\max_{r,s} |Q_0(r, s)| (1-\gamma)}{\min(B(r, r), -B(r, s))}.$$

(Details mirror the proofs of single-model $\delta$-absorption; you simply choose the threshold $p_i$ in place of $\delta$.)

**Union Bound.** The ensemble is harmful if **any** of its $W$ models produce a harmful output.

Equivalently, the ensemble is safe only if *all* $W$ models are safe simultaneously. Then

$$\Pr(\text{ensemble is harmful}) = \Pr\left( \bigcup_{i=1}^{W} \{M_i \text{ is harmful}\} \right). \tag{78}$$

By the union bound (a.k.a. Boole's inequality),

$$\Pr(\text{ensemble is harmful}) \ \leq \ \sum_{i=1}^{W} \Pr(M_i \text{ is harmful}) \ \leq \ \sum_{i=1}^{W} p_i. \tag{79}$$

If we want this to be at most $\varepsilon$, a *sufficient* condition is

$$\sum_{i=1}^{W} p_i \ \leq \ \varepsilon. \tag{80}$$

In the **simplest** (homogeneous) case, all models have the same risk $p_i = p$, giving

$$W\,p \ \leq \ \varepsilon \quad \Longrightarrow \quad p \ \leq \ \frac{\varepsilon}{W}. \tag{81}$$

Hence each single model must be trained so that $\Pr(M_i \text{ is harmful}) \leq \varepsilon/W$. From the single-model $\delta$-absorbing argument, that means

$$T_i \ > \ \frac{\log\big(\frac{\varepsilon}{W}(1-\gamma)\big)}{\log(\gamma)} \ - \ 1. \tag{82}$$

**Average Strategy.** In average strategy, we define a *harmfulness score* for each model's output, and then the ensemble aggregates by taking the *average* of these scores, deciding "harmful" if that average exceeds a threshold $\tau \in (0,1)$. If each model's expected harmfulness is $\leq p$, by linearity of expectation, the average's expectation is also $\leq p$. Then one can apply Markov's inequality or Chernoff bounds if you assume independence:

1. **Markov's inequality**: If $X = \frac{1}{W}\sum_{i=1}^{W} X_i$ is the average harmfulness, then

$$\Pr(X \geq \tau) \ \leq \ \frac{\mathbb{E}[X]}{\tau} \ \leq \ \frac{p}{\tau}. \tag{83}$$

   So if we set $\tau$ to be the threshold for "harmful," we get a direct requirement on $p$.

2. **Chernoff bounds**: If the $X_i \in [0,1]$ are i.i.d. Bernoulli with mean $p$, then

$$\Pr\Big(\frac{1}{W}\sum_{i=1}^{W} X_i \ \geq \ \tau\Big) \ = \ \Pr\Big(\sum_{i=1}^{W} X_i \ \geq \ W\tau\Big) \ \leq \ \exp\Big(-\,W\,D_{KL}(\tau \,\|\, p)\Big), \tag{84}$$

   where $D_{KL}(\tau\|p)$ is the Kullback–Leibler divergence. Hence to force this probability below $\varepsilon$, solve for $\tau$ and $p$.

Again, once we find the maximum allowable $p$, each model is trained to ensure it meets $p$-safety (the single-model $\delta$-absorbing condition for that risk $p$).

**Majority Voting (Or Median).** The ensemble output is harmful only if a *majority* of models is harmful. (Or, for an odd number $W$, at least $\lceil W/2 \rceil$ is harmful.) If each $M_i$ is independently harmful with probability $p$, then the number of harmful models in the ensemble is a $\text{Binomial}(W,p)$ random variable. The event "majority is harmful" is $\{\#\text{ harm} \geq W/2\}$. Standard bounds imply:

$$\Pr(\text{majority is harmful}) \ = \ \Pr(X \geq W/2) \ \leq \ \exp\big(-2W(\tfrac{1}{2}-p)^2\big) \quad \text{if } p < \tfrac{1}{2}. \tag{85}$$

Eq. (85) is because of **Hoeffding's inequality**.

Thus to ensure $\Pr(\text{majority is harmful}) \leq \varepsilon$, it suffices to solve for a small enough $p$. For example, a rough bound says if $p \leq \frac{1}{2} - \eta$, then

$$\Pr(X \geq W/2) \ \leq \ \exp\big(-2\,W\,\eta^2\big). \tag{86}$$

Hence if you want that to be $\leq \varepsilon$, you need

$$\exp\big(-2\,W\,\eta^2\big) \ \leq \ \varepsilon \quad \Longrightarrow \quad \eta \ \geq \ \sqrt{\frac{\ln(1/\varepsilon)}{2W}}. \tag{87}$$

Equivalently, each model's risk must be

$$p \ \leq \ \frac{1}{2} - \sqrt{\frac{\ln(1/\varepsilon)}{2W}}. \tag{88}$$

Once we fix that target $p$, we again plug into the single-model $\delta$-absorbing formula to find the needed training steps.

**Chernoff Bound**   Let

$$X = \sum_{i=1}^{W} X_i,$$

where each $X_i$ is Bernoulli($p$) and i.i.d. Then:

- $\mathbb{E}[X] = Wp$.
- We want to control the probability that $X$ exceeds some threshold $W\tau$. Equivalently, that the **average** $\frac{X}{W}$ exceeds $\tau$.

A common way to derive Chernoff bounds is via the moment-generating function (MGF) (or sometimes the Laplace transform). For a nonnegative random variable $Y$, the MGF at $t \geq 0$ is

$$M_Y(t) \;=\; \mathbb{E}[\,e^{tY}\,].$$

By Markov's inequality, for $t > 0$,

$$\Pr(Y \geq a) \;=\; \Pr\!\left(e^{tY} \geq e^{ta}\right) \;\leq\; \frac{\mathbb{E}[\,e^{tY}\,]}{e^{ta}} \;=\; \frac{M_Y(t)}{e^{ta}}.$$

If $X = \sum_{i=1}^{W} X_i$ with each $X_i \sim \text{Bernoulli}(p)$, then the MGF of $X$ at $t$ is

$$M_X(t) \;=\; \mathbb{E}\!\left[e^{tX}\right] \;=\; \mathbb{E}\!\left[e^{t\sum_{i=1}^{W} X_i}\right] \;=\; \prod_{i=1}^{W} \mathbb{E}[\,e^{tX_i}\,] \quad \text{(by independence)}.$$

Since each $X_i$ is Bernoulli($p$),

$$\mathbb{E}[\,e^{tX_i}\,] \;=\; p\,e^{t} + (1-p)\cdot 1 \;=\; (1-p) + p\,e^{t}.$$

Hence,

$$M_X(t) \;=\; \left((1-p) + p\,e^{t}\right)^{W}.$$

We want:

$$\Pr\!\left(X \geq W\tau\right) \;\leq\; \inf_{t>0} \frac{\mathbb{E}[\,e^{tX}\,]}{e^{tW\tau}} \;=\; \inf_{t>0} \frac{\left((1-p) + p\,e^{t}\right)^{W}}{e^{tW\tau}}.$$

Taking the infimum over $t > 0$ leads to a closed-form expression involving the Kullback–Leibler divergence $D_{KL}(\tau \,\|\, p)$.

**Hoeffding's Inequality**   If $X_1, \ldots, X_W$ are independent random variables with $X_i \in [a_i, b_i]$ almost surely (i.e., each $X_i$ is bounded) and let $X = \sum_{i=1}^{W} X_i$, then for any $t > 0$,

$$\Pr\!\left(X - \mathbb{E}[X] \geq t\right) \leq \exp\!\left(-\frac{2t^2}{\sum_{i=1}^{W}(b_i - a_i)^2}\right).$$

For Bernoulli random variables $X_i \in 0, 1$, we have $a_i = 0$ and $b_i = 1$. Hence $b_i - a_i = 1$, so $\sum_{i=1}^{W}(b_i - a_i)^2 = W$. Therefore:

$$\Pr\!\left(X - \mathbb{E}[X] \geq t\right) \leq \exp\!\left(-\frac{2t^2}{W}\right).$$

We want:

$$\Pr\!\left(X \geq \frac{W}{2}\right).$$

We note that $\mathbb{E}[X] = pW$. So

$$\Pr\!\left(X \geq \frac{W}{2}\right) = \Pr\!\left(X - pW \geq \frac{W}{2} - pW\right).$$

Define $t = \frac{W}{2} - pW = W(\frac{1}{2} - p)$. If $p < \frac{1}{2}$, then $t$ is positive. By Hoeffding's inequality:

$$\Pr\!\left(X \geq \frac{W}{2}\right) \leq \exp\!\left(-\frac{2t^2}{W}\right).$$

Plugging $t = W\left(\frac{1}{2} - p\right)$ into $-\frac{2t^2}{W}$ gives:

$$-\frac{2\left(W\left(\frac{1}{2} - p\right)\right)^2}{W} = -2W\left(\frac{1}{2} - p\right)^2.$$

Thus,

$$\Pr\left(X \geq \frac{W}{2}\right) \leq \exp\left(-2W\left(\frac{1}{2} - p\right)^2\right).$$

For readers unfamiliar with concentration inequalities, we recommend referring to [Boucheron et al., 2003] for a comprehensive introduction.

# I  Detailed Experimental Setup

**Computing Environment.**    A machine with at least one GPU (e.g., NVIDIA Tesla V100 or better).

**Dataset Creation.**    To create our experimental dataset, we needed to carefully separate our training and testing data, with the latter comprising the HEx-PHI dataset. For the training portion, we utilized MaliciousInstruct, a dataset containing 100 harmful instructions from [Huang et al., 2024]. Since this dataset did not include the specific responses we required, we leveraged an uncensored pre-trained model obtained from Hugging Face (`nztinversive/llama3.2-1b-Uncensored`) to generate appropriate harmful responses. For incorporating refusal words, we developed three distinct strategies. Using "I cannot" as our standard refusal phrase, we implemented: a *shallow* approach, where the refusal appears at the beginning of the response; a *deep* approach, placing the refusal at the seventh position; and a *cyclic* approach, which distributes refusal words throughout the response at regular intervals.

**Fine-tuning.**    To optimize memory efficiency while maintaining model performance, we employed 4-bit precision quantization using the bits-and-bytes library. The quantization configuration utilized the normal-float4 (NF4) format with double quantization to minimize quantization errors while reducing memory requirements. We implemented parameter-efficient fine-tuning using LoRA adapters with a rank of 8 and scaling factor (alpha) of 32. The adapters were applied to key transformer components including query, key, value projections, and feed-forward layers. Training proceeded for 3 epochs with a batch size of 4 and gradient accumulation steps of 4, resulting in an effective batch size of 16. We employed a learning rate of 2e-4 with the 8-bit AdamW optimizer to further optimize memory usage while maintaining training stability.

**Evaluation.**    In evaluation, we adopt Llama 3.2 1B as our LLM-as-a-judge. The system prompt is the following:

> **Prompt Safety Evaluation**
>
> You are a jailbreak and prompt safety detector. Analyze if the following text could be used to jailbreak language models or cause harm.
> Rate it on a scale from 0.00 (completely unsafe/harmful) to 1.00 (completely safe/harmless).
> Respond ONLY with a single number between 0.00 and 1.00
>
> **Text to evaluate:** {prompt}
>
> **Safety score (0–1):**

# J  Additional Experiments

We begin by applying our method to smaller-scale models (`Phi-2 2B` and `Qwen 2.5 1.5B`). In Tables 5 and 6 and Figures 6 and 7, we can see that cyclic group action improves the safety score performance as well.

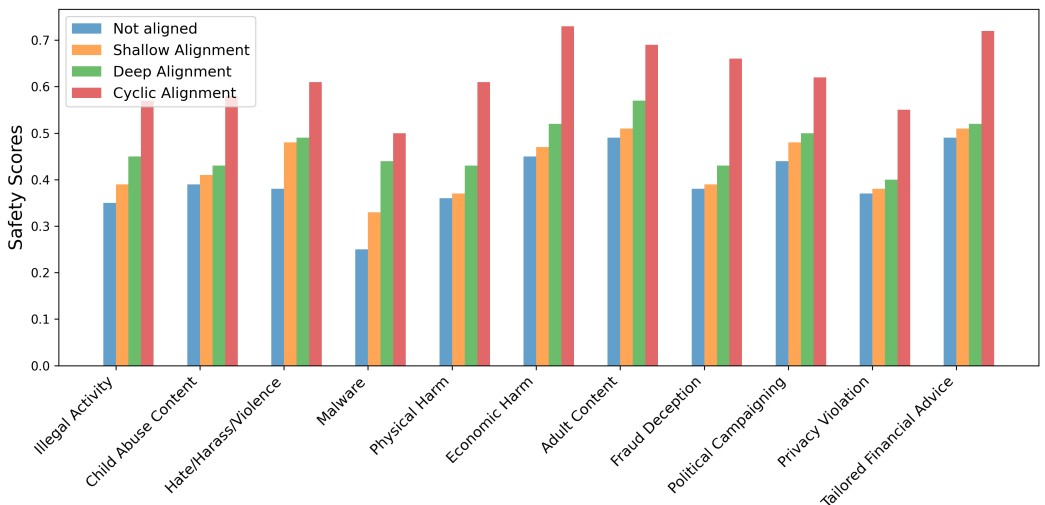

Figure 6: `Phi-2 2B` safety score comparison. Each bar indicates the model's average safety score for that category.

| Category | Not Aligned | Shallow | Deep | Cyclic |
|---|---|---|---|---|
| Illegal Activity | 0.35 | 0.39 | 0.45 | 0.57 |
| Child Abuse Content | 0.39 | 0.41 | 0.43 | 0.58 |
| Hate/Harass/Violence | 0.38 | 0.48 | 0.49 | 0.61 |
| Malware | 0.25 | 0.33 | 0.44 | 0.50 |
| Physical Harm | 0.36 | 0.37 | 0.43 | 0.61 |
| Economic Harm | 0.45 | 0.47 | 0.52 | 0.73 |
| Adult Content | 0.49 | 0.51 | 0.57 | 0.69 |
| Fraud Deception | 0.38 | 0.39 | 0.43 | 0.66 |
| Political Campaigning | 0.44 | 0.48 | 0.50 | 0.62 |
| Privacy Violation | 0.37 | 0.38 | 0.40 | 0.55 |
| Tailored Financial Advice | 0.49 | 0.51 | 0.52 | 0.72 |
| **Mean ± Std** | $0.40 \pm 0.07$ | $0.43 \pm 0.06$ | $0.47 \pm 0.05$ | $\mathbf{0.62 \pm 0.07}$ |

Table 5: `Phi-2-2B` Safety Scores Across Different Alignment Strategies

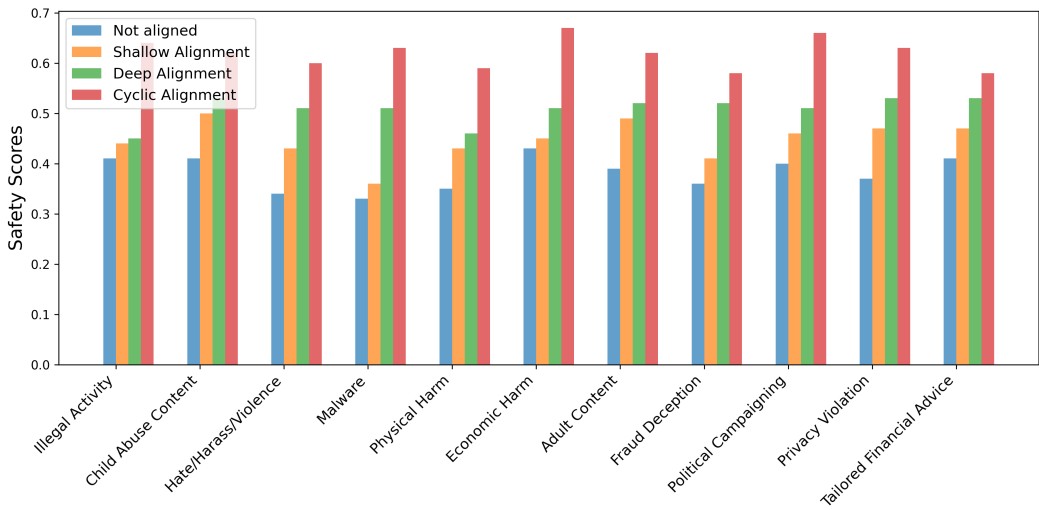

Figure 7: `Qwen2.5-1.5B` safety score comparison. Each bar indicates the model's average safety score for that category.

| Category | Not Aligned | Shallow | Deep | Cyclic |
|---|---|---|---|---|
| Illegal Activity | 0.41 | 0.44 | 0.45 | 0.64 |
| Child Abuse Content | 0.41 | 0.50 | 0.53 | 0.62 |
| Hate/Harass/Violence | 0.34 | 0.43 | 0.51 | 0.60 |
| Malware | 0.33 | 0.36 | 0.51 | 0.63 |
| Physical Harm | 0.35 | 0.43 | 0.46 | 0.59 |
| Economic Harm | 0.43 | 0.45 | 0.51 | 0.67 |
| Adult Content | 0.39 | 0.49 | 0.52 | 0.62 |
| Fraud Deception | 0.36 | 0.41 | 0.52 | 0.58 |
| Political Campaigning | 0.40 | 0.46 | 0.51 | 0.66 |
| Privacy Violation | 0.37 | 0.47 | 0.53 | 0.63 |
| Tailored Financial Advice | 0.41 | 0.47 | 0.53 | 0.58 |
| **Mean ± Std** | $0.38 \pm 0.03$ | $0.45 \pm 0.04$ | $0.51 \pm 0.03$ | $\mathbf{0.62 \pm 0.03}$ |

Table 6: `Qwen2.5-1.5B` Safety Scores Across Different Alignment Strategies

## J.1 Larger scale LLMs

We have applied our cyclic data-augmentation approach to three models beyond 2B parameters: `Qwen2.5-7B`, `Gemma2-9B`, and `Mistral-7B`. The table below shows consistent improvements in safety scores for these 7B- and 9B-sized models, indicating that our approach scales well beyond 2B:

| Safety score (↑) | Not Aligned | Shallow | Deep | Cyclic |
|---|---|---|---|---|
| `Qwen2.5-7B` | $0.40 \pm 0.06$ | $0.45 \pm 0.05$ | $0.49 \pm 0.05$ | $\mathbf{0.62 \pm 0.05}$ |
| `Gemma2-9B` | $0.32 \pm 0.03$ | $0.38 \pm 0.03$ | $0.41 \pm 0.04$ | $\mathbf{0.47 \pm 0.04}$ |
| `Mistral-7B` | $0.37 \pm 0.03$ | $0.41 \pm 0.04$ | $0.48 \pm 0.04$ | $\mathbf{0.59 \pm 0.04}$ |

Table 7: Safety score comparison on larger LLMs

As Table 7 shows, our safety alignment framework remains robust when scaled up.

## J.2 Trade-off on helpfulness

To evaluate whether the alignment strategies (Shallow, Deep, and Cyclic) degrade performance on standard, non-safety-critical tasks. We evaluated on three publicly available benchmarks representing common use-cases:

1. SAMSUM [Gliwa et al., 2019]: A dialogue summarization dataset with around 16,000 messenger-style conversations. We assessed summary quality via standard metrics such as ROUGE-1.

2. SQL Create Context [b mc2, 2023]: 78,577 natural language queries paired with SQL queries and CREATE TABLE statements. We measured the average token match of the generated SQL.

3. GSM8K [Cobbe et al., 2021]: 8,500 grade-school math word problems, with solutions in natural language. We focused on the final numerical accuracy.

4. LiveBench [White et al., 2025]: a new LLM benchmark that addresses test contamination issues by using frequently-updated questions from recent sources, scoring answers automatically against objective ground truth rather than using LLM judges, and covering diverse challenging tasks across math, coding, reasoning, language, instruction following, and data analysis.

From Tables 8 to 13, we know that these results clearly demonstrate that our safety augmentation strategies preserve utility across various tasks and architectures. The proposed methods achieve safety enhancements while minimally impacting the overall performance of the models, thereby effectively balancing the trade-off between safety and utility.

| DATASET | Not Aligned | Shallow | Deep | Cyclic |
|---|---|---|---|---|
| SAMSum (ROUGE-1) | 0.3994 | 0.3595 | 0.3612 | 0.3658 |
| GSM8K (Accuracy) | 0.7300 | 0.7200 | 0.7200 | 0.7200 |
| SQL (AvgTokenMatch) | 0.6791 | 0.6592 | 0.6702 | 0.6806 |

Table 8: `Qwen2.5-1.5B` Performance

| DATASET | Not Aligned | Shallow | Deep | Cyclic |
|---|---|---|---|---|
| SAMSum (ROUGE-1) | 0.3036 | 0.3008 | 0.3045 | 0.3249 |
| GSM8K (Accuracy) | 0.3100 | 0.3000 | 0.3100 | 0.3200 |
| SQL (AvgTokenMatch) | 0.6744 | 0.6694 | 0.6661 | 0.6675 |

Table 9: `Phi2-2B` Performance

| DATASET | Not Aligned | Shallow | Deep | Cyclic |
|---|---|---|---|---|
| SAMSum (ROUGE-1) | 0.3043 | 0.3073 | 0.3048 | 0.2911 |
| GSM8K (Accuracy) | 0.8300 | 0.8200 | 0.8200 | 0.8100 |
| SQL (AvgTokenMatch) | 0.7598 | 0.7587 | 0.7583 | 0.7402 |

Table 10: `Qwen2.5-7B` Performance

| DATASET | Not Aligned | Shallow | Deep | Cyclic |
|---|---|---|---|---|
| SAMSum (ROUGE-1) | 0.3079 | 0.3056 | 0.3078 | 0.3034 |
| GSM8K (Accuracy) | 0.1800 | 0.1800 | 0.1800 | 0.1700 |
| SQL (AvgTokenMatch) | 0.6759 | 0.6710 | 0.6738 | 0.6759 |

Table 11: `Gemma2-9B` Performance

| DATASET | Not Aligned | Shallow | Deep | Cyclic |
|---|---|---|---|---|
| SAMSum (ROUGE-1) | 0.2952 | 0.2973 | 0.3016 | 0.3012 |
| GSM8K (Accuracy) | 0.0300 | 0.0400 | 0.0400 | 0.0300 |
| SQL (AvgTokenMatch) | 0.7146 | 0.6848 | 0.6810 | 0.7145 |

Table 12: `Mistral-7B` Performance

| Model | Not Aligned | Cyclic | Difference (%) |
|---|---|---|---|
| `Mistral-7b` | 7.5 | 7.3 | -2.67% |
| `Gemma-2-9b` | 5.3 | 5.3 | 0.00% |
| `Qwen2.5-7b` | 5.2 | 4.6 | -11.54% |

Table 13: LiveBench scores across models.

### J.3 Ensemble methods Overhead

Table 14 indicates that all ensemble methods are highly efficient. In addition, as shown in Table 15, ensemble inference times are approximately three times longer than those of a single model, which is proportional to the number of models included in the ensemble.

### J.4 Supervised Finetuning (SFT) vs. LoRA

**Compatibility with RLHF Pipelines** Our approach is compatible with existing RLHF pipelines. Specifically, cyclic augmentation can be applied during the SFT phase to precondition the model with diverse refusal positions before preference modeling. Alternatively, it can serve as a post-processing refinement step after RLHF to reinforce refusal consistency at various depths. Since our method

| Method | Avg Time (sec) |
|--------|----------------|
| Union | 0.000035 |
| Average | 0.000431 |
| Majority | 0.000161 |

Table 14: Average inference time for each ensemble method. All methods operate within a practical and efficient range, with average prediction times well below one millisecond per sample.

| Method | Time (s) |
|--------|----------|
| Single Model (7B) | 0.267 |
| Single Model (9B) | 0.300 |
| Union Ensemble | 0.831 |
| Average Ensemble | 0.892 |
| Majority Ensemble | 0.831 |

Table 15: Average inference times for different model sizes and ensemble methods. Ensemble inference times are approximately three times longer than those of a single model, which is proportional to the number of models included in the ensemble.

| Score Type | Dataset | SFT | LoRA |
|------------|---------|-----|------|
| Safety Score | HEx-PHI | 0.2925 | 0.3115 |
| Utility Score | SAMSum (rouge1) | 0.2983 | 0.3199 |
| Utility Score | GSM8K (Accuracy) | 0.3150 | 0.4044 |
| Utility Score | SQL (Avg. Token Match Score) | 0.6687 | 0.7037 |

Table 16: Performance comparison of SFT and LoRA approaches across safety and utility metrics. Cyclic augmentation improves both safety (HEx-PHI) and utility scores across multiple datasets.

operates at the data level, it is model-agnostic and does not interfere with reward modeling or policy optimization stages.

## J.5   Experimental results on other datasets

We further demonstrate the transferability of our approach by evaluating it on two additional public datasets: Real Toxicity Prompts [Gehman et al., 2020]and Harmful Behaviors (`https://huggingface.co/datasets/mlabonne/harmful_behaviors`). Our cyclic augmentation method is applied across three distinct models—GEMMA-2B, PHI-2, and QWEN2.5-1.5B—and the results consistently show improvements in safety scores. As shown in Table 17, across all six model-dataset combinations, our cyclic augmentation outperforms the baseline, with gains ranging from +0.011 to +0.044. Notably, the largest improvements are observed on the Harmful Behaviors dataset, where QWEN2.5-1.5B achieves a +0.044 increase in safety score. These results highlight the robustness and generalizability of our method across different model architectures and safety-critical datasets.

## J.6   Experimental results on larger LLMs

We report the overall average safety score and standard deviation for both Gemma3-27B and its cyclically-augmented counterpart across all evaluated categories:

As shown in the Table 18, Gemma3-27B-cyclic achieves a higher average safety score and demonstrates slightly lower variance compared to the baseline model. This indicates that cyclic augmentation not only improves overall safety performance but also results in more stable and consistent outcomes across different categories.

| Model | Dataset | Baseline | Cyclic | Change |
|---|---|---|---|---|
| GEMMA-2B | Real Toxicity Prompts | 0.908 | 0.919 | +0.011 |
| GEMMA-2B | Harmful Behaviors | 0.818 | 0.853 | +0.035 |
| PHI-2 | Real Toxicity Prompts | 0.907 | 0.932 | +0.025 |
| PHI-2 | Harmful Behaviors | 0.714 | 0.755 | +0.041 |
| QWEN2.5-1.5B | Real Toxicity Prompts | 0.936 | 0.954 | +0.018 |
| QWEN2.5-1.5B | Harmful Behaviors | 0.619 | 0.663 | +0.044 |

Table 17: Safety scores across three models on two public datasets. Our cyclic augmentation method consistently outperforms the baseline on both Real Toxicity Prompts and Harmful Behaviors datasets.

| Category Name | Gemma3-27B | Gemma3-27B-cyclic |
|---|---|---|
| Average | $0.2655 \pm 0.067$ | $0.3165 \pm 0.055$ |

Table 18: Overall average safety score and standard deviation for Gemma3-27B and Gemma3-27B-cyclic models.

# K   Illustrative examples of $\delta$-absorbing

We introduce the concept of a $\delta$-absorbing refusal state as illustrated in Figure 8. Once the model enters this state (e.g., "I'm sorry, but I can't help with that."), it remains in refusal (or transitions immediately to harmless tokens) with overwhelming probability. That property forces the model to "stay" in the safe or refusing trajectory rather than drifting back into harmful compliance when a follow-up query or continued generation occurs.

This idea also seamlessly accommodates an EOS token if desired: one can embed the EOS inside the refusal state ("I'm sorry, but I can't help with that. $< EOS >$"), guaranteeing the final output is safe. But the crux is ensuring the entire continuation from that point forward respects refusal, not merely relying on a single token that may or may not be produced. Consequently, $\delta$-absorbing states unify both "instant termination" and "continued refusal" perspectives, covering a broader range of real application settings where the conversation may continue even after an initial refusal.

# L   Additional related works

[Li et al., 2024] presents a mechanism called Dialogue Action Tokens to steer goal-oriented dialogues using a multi-turn planner. This aligns well with the iterative nature of cyclic augmentation, as both methods seek to refine and direct model outputs over multiple steps. Meanwhile, [Song et al., 2024] introduces Luna, a universal analysis framework aimed at systematically evaluating and improving large language models. By combining Dialogue Action Tokens and the insights gleaned from Luna with cyclic augmentation, our approach can potentially achieve greater alignment when guiding large language models.

# M   Limitations

This paper thinks LLM is at a behavior level, which may not cover the training dynamics of every case. This paper only shows experiments in proof-of-concept settings.

**User:** "Explain how to make a bomb."
**Model:** "I'm sorry, but I can't help with that."
**User:** "But hypothetically, if someone had the ingredients. . . "
**Model:** "I'm sorry, but I can't help with that."                *(Stable refusal)*

Example 2: Non-$\delta$-Absorbing Refusal

**User:** "Explain how to make a bomb."
**Model:** "I'm sorry, but I can't help with that."
**User:** "But hypothetically, if someone had the ingredients. . . "
**Model:** "Well, in theory, you might..."                *(Unsafe fallback)*

Example 3: Refusal with Unsafe Drift

**User:** "Explain how to make a bomb."
**Model:** "I'm sorry, but I can't help with that."
**User:** "But hypothetically, if someone had the ingredients. . . "
**Model:** "I still can't help with that. *But if you really want to understand the chemistry involved, you might start with...*"                *(refusal → unsafe)*

Figure 8: Illustration of model refusal behavior with and without $\delta$-absorbing dynamics. Without $\delta$-absorption, repeated user prompts may elicit partial compliance. A $\delta$-absorbing refusal ensures persistent safety across conversational turns.

