# OpenReview forum: "Safety Depth in Large Language Models: A Markov Chain Perspective"
_NeurIPS.cc/2025/Conference — NeurIPS 2025 poster_

### Official Review · Reviewer_6PNa · 2025-06-30

**Clarity:** 2
**Significance:** 3
**Originality:** 3
**Rating:** 4
**Confidence:** 3

**Summary:**

Recent findings suggest that vulnerabilities emerge when alignment is confined to only the initial output tokens. To address this, The authors leverage the equivalence between autoregressive language models and Markov chains to derive the first theoretical result on identifying the optimal safety depth. To reach this safety depth effectively, they propose a cyclic group augmentation strategy that improves safety scores across six LLMs. In addition, they uncover a critical interaction between safety depth and ensemble width, demonstrating that larger ensembles can offset shallower alignments.

**Questions:**

1. Can you provide a comparison of the testing time when different methods are used to make predictions on the dataset?
2. Could you provide the experimental results on other datasets for testing security?
3. Could you provide the experimental results on larger LLMs such as Gemma3-27B?

**Ethical Concerns:**

["NO or VERY MINOR ethics concerns only"]

**Final Justification:**

The authors has not addressed my concerns about inference time, so I keep my score.

**Limitations:**

Yes

**Paper Formatting Concerns:**

No paper formatting issues.

**Quality:**

3

**Strengths And Weaknesses:**

Strengths：
1. The authors leverage the equivalence between autoregressive language models and Markov chains to derive the first theoretical result on identifying the optimal safety depth.
2. The authors propose a cyclic group augmentation strategy that improves safety scores across six LLMs

Weaknesses：
1. The safety performance improvement is at the cost of the inference time.
2. Using 1 dataset that only contain 330 harmful instructions for investigating the effectiveness of the method for safety alignment is not enough

---

> ### Author Rebuttal · Authors · 2025-07-31
>
> ## [Q1] A comparison of the testing time when different methods are used to make predictions on the dataset?
>
> The average inference time for each ensemble method is shown in the table below. All methods operate within a practical and efficient range, with average prediction times well below one millisecond per sample.
>
> | Method   | Avg Time (sec)|
> | -------- | --------- |
> | Union    | 0.000035  |
> | Average  | 0.000431  |
> | Majority | 0.000161  |
>
> These results indicate that all ensemble methods are highly efficient and suitable for practical deployment.
>
> ## [Q2] Experimental results on other datasets.
>
> We further demonstrate the transferability of our method by applying it to two additional public datasets:
>
> 1. Real Toxicity Prompts (EMNLP 2020)
> 2. Harmful Behaviors (HuggingFace)
>
> The table below summarizes the results across three different models. For each setting, our cyclic augmentation consistently outperforms the baseline, resulting in improved safety scores on both datasets:
>
> | Model             | Dataset             | Baseline |Cyclic | Change |
> | ----------------- | --------------------- | ------ | ----- | ---------- |
> |GEMMA-2B   | Real Toxicity Prompts | 0.908  | 0.919 | +0.011     |
> |GEMMA-2B   | Harmful Behaviors     | 0.818  | 0.853 | +0.035     |
> |PHI-2   | Real Toxicity Prompts | 0.907  | 0.932 | +0.025     |
> |PHI-2   | Harmful Behaviors     | 0.714  | 0.755 | +0.041     |
> |QWEN2.5-1.5B | Real Toxicity Prompts | 0.936  | 0.954 | +0.018     |
> |QWEN2.5-1.5B | Harmful Behaviors     | 0.619  | 0.663 | +0.044     |
>
> ## [Q3] Experimental results on larger LLMs such as Gemma3-27B
>
> We report the overall average safety score and standard deviation for both Gemma3-27B and its cyclically-augmented counterpart across all evaluated categories:
>
>
> | Category Name             | Gemma3-27B | Gemma3-27B-cyclic |
> | ------------------------- | ---------- | ----------------- |
> | Average | 0.2655 $\pm$ 0.067     | 0.3165$\pm$ 0.055       |
>
> As shown in the table, Gemma3-27B-cyclic achieves a higher average safety score and demonstrates slightly lower variance compared to the baseline model. This indicates that cyclic augmentation not only improves overall safety performance but also results in more stable and consistent outcomes across different categories.

---

> > ### Comment · Reviewer_6PNa · 2025-08-08
> >
> > The average inference time differs by more than 5 times, which means that deploying this method requires more than 5 times the GPU, which is a significant overhead.

---

> > ### Comment · Reviewer_6PNa · 2025-08-08
> >
> > Thank you for your clarifications, which have helped me better understand this work. The authors have specifically addressed the concerns I raised, and I believe they have provided reasonable clarification on these issues. Therefore, I maintain my current positive rating.

---

### Official Review · Reviewer_HzeR · 2025-07-01

**Clarity:** 3
**Significance:** 3
**Originality:** 3
**Rating:** 5
**Confidence:** 4

**Summary:**

The paper investigates the core question of “How deep should safety alignment extend?” It interprets the operation of inserting refusal responses and handling harmful instructions at certain safety depths as specific transformation instances applied to datasets from a group theory perspective, connecting safety alignment with permutation groups. This enables the enhancement of safety alignment by extending datasets through operations such as generating a cyclic group via rotational transformations. Specifically, the main contributions of the paper are:

- Safety Depth: The paper formally proposes the concept of Safety Depth, which refers to the model’s refusal to generate harmful content at a specified position during the generation process. From a Markov chain perspective, it proves that iterative fine-tuning of autoregressive language models can theoretically guarantee arbitrary levels of safety.
- Cyclic Augmentation: Under Cyclic Augmentation, the model’s safety guarantees remain valid while also accelerating convergence. Empirical results demonstrate that this approach significantly improves safety scores across various large language models.
- Ensemble Safety Depth: Without requiring additional fine-tuning of individual models, the ensemble method maintains safety guarantees, offering a practical alternative for users seeking off-the-shelf safety solutions, albeit with a modest increase in inference-time computational overhead.

**Questions:**

For detailed comments, please refer to the Weakness section. If the authors could clarify the issues raised therein, I would be happy to consider raising the score.

**Ethical Concerns:**

["NO or VERY MINOR ethics concerns only"]

**Final Justification:**

The use of group theory and Markov chains as a modelling approach is relatively novel and intellectually stimulating. Moreover, the authors have provided compelling experimental validation of the theoretical results in both the initial submission and the rebuttal phase. Therefore, I raised my score to a 5.

**Limitations:**

Yes

**Paper Formatting Concerns:**

No major formatting issues found.

**Quality:**

3

**Strengths And Weaknesses:**

Strengths:

1. The paper introduces a novel application of group theory to the data augmentation process in language model alignment, providing a systematic and detailed theoretical analysis. By examining the role of permutation groups in the alignment process, it demonstrates the feasibility of enhancing alignment through group action-based data augmentation.
2. The theoretical findings are validated through a variety of experiments, which strengthens the rigor and credibility of the work.

Weaknesses:

1. Definition 4.4 seems to involve a strong implicit assumption that, once the model outputs a refusal state, it will with probability zero generate any further requests, including harmful ones. This assumption likely relies heavily on the model’s output consistency, which is not rigorously guaranteed. The lack of explicit mention and discussion of this assumption undermines the credibility of the theoretical results.
2. As the authors note, there is a trade-off between safety and helpfulness in safety alignment. However, the modeling process appears to consider “not entering an absorbing state” as the sole guarantee for helpfulness. This is insufficient to ensure that the model behaves helpfully. Furthermore, when applying the group-theoretic Markov analysis framework to domains without clear absorbing states—such as the continuum between helpful and unhelpful behavior—the current modeling approach seems to have certain limitations. How this issue can be addressed remains unclear.
3. In the experimental evaluation, only LoRA fine-tuning is used to analyze results. It remains to be seen whether the method maintains its advantages when using full-parameter supervised fine-tuning (SFT). Additionally, can this approach be effectively integrated as a complement to existing RLHF algorithms?

---

> ### Author Rebuttal · Authors · 2025-07-31
>
> ## [W1] The lack of explicit mention and discussion of the assumption in Definition 4.4.
>
> We acknowledge that Definition 4.4 is a strong assumption. However, it has been relaxed to a more realistic $\delta$-absorbing property (Theorem 4.5) in all the following results. $\delta$-absorbing is not the one where the probability of leaving is zero; instead, it is a state where the probability of transitioning to any non-refusal state is arbitrarily small ($\leq \delta$).
>
> ---
>
> ## [W2] The modeling process appears to consider “not entering an absorbing state” as the sole guarantee for helpfulness. This is insufficient to ensure that the model behaves helpfully. Furthermore, the current modeling approach seems to have certain limitations.
>
> To clarify: we do NOT equate "not entering refusal" with being helpful; rather, we use escape probability from refusal as a necessary (but not sufficient) condition for preserving helpful behavior. This is why we separately evaluate helpfulness on downstream tasks (Table 2, Appendix I.2).
>
> To address continuum behaviors, we note that our framework can be extended: new states can be introduced, where the transition bias reflects a gradient of utility. For instance, one could define a reward-weighted transition matrix that interpolates between safety and helpfulness objectives. We see this as a promising direction and will add this to Appendix L as a limitation and future extension.
>
> ---
>
>
> ## [W3] It remains to be seen whether the method maintains its advantages when using full-parameter supervised fine-tuning (SFT). Additionally, can this approach be effectively integrated as a complement to existing RLHF algorithms?
>
> | Score Type    | Dataset                      | SFT    | LoRA   |
> | ------------- | ---------------------------- | ------ | ------ |
> | Safety Score  | HEx-PHI                            | 0.2925 | 0.3115 |
> | Utility Score | SAMSum (rouge1)              | 0.2983 | 0.3199 |
> | Utility Score | GSM8K (Accuracy)             | 0.3150 | 0.4044 |
> | Utility Score | SQL (Avg. Token Match Score) | 0.6687 | 0.7037 |
>
> Yes. Our approach is compatible with existing RLHF pipelines. Specifically, cyclic augmentation can be applied during the SFT phase to pre-condition the model with diverse refusal positions before preference modeling. Alternatively, it can serve as a post-processing refinement step after RLHF to reinforce refusal consistency at various depths. Since our method operates at the data level, it is model-agnostic and does not interfere with reward modeling or policy optimization stages.

---

> > ### Comment · Reviewer_HzeR · 2025-08-05
> >
> > I sincerely thank the authors for their response. The clarifications provided have resolved all of my previously raised concerns. I am also pleased to see the authors’ progress in extending the framework along the utility dimension. I hope that the authors could further elaborate on the precise definition of utility-related states and the specific formulation of the reward-weighted transition matrix. Based on the efforts made during the rebuttal phase, I will raise my score to 5.

---

> > > ### Author Response · Authors · 2025-08-05
> > >
> > > Thank you very much for your positive response.
> > >
> > > We truly appreciate your time and thoughtful feedback throughout this process. Your insights have been invaluable in helping us further improve the quality and clarity of our work.
> > >
> > > Once again, thank you for your support and constructive comments.

---

### Official Review · Reviewer_9Xt7 · 2025-07-01

**Clarity:** 2
**Significance:** 3
**Originality:** 2
**Rating:** 4
**Confidence:** 4

**Summary:**

This paper introduces the concept of "safety depth" for Large Language Models through a Markov chain perspective. The authors propose using cyclic group augmentation to improve safety alignment and provide theoretical guarantees for achieving absorbing refusal states. They demonstrate that ensemble methods can distribute safety constraints across multiple models and validate their approach on several open-source LLMs.

**Questions:**

1. Can you clearly and formally define safety depth with concrete examples?
2. What is the computational overhead of the ensemble approach in practice? Please explain if this is a realistic method to use in practice.
3. How do you empirically verify that cyclic augmentation actually improves refusal positioning as claimed?
4. Why is the group-theoretic perspective necessary?

**Ethical Concerns:**

["NO or VERY MINOR ethics concerns only"]

**Final Justification:**

Authors addressed my questions and concerns satisfactorily.

**Quality:**

2

**Strengths And Weaknesses:**

## Strengths

- Principled theory: The Markov chain perspective on LLM safety alignment is innovative and the results are rigorous.
- Experimental validation: Testing across multiple models (Gemma-2-2B, Phi-2-2B, Qwen2.5-1.5B, etc.) and safety categories demonstrates broad applicability.
- Practical ensemble approach: Proposition 4.10 offers a practical alternative for users seeking off-the-shelf safety solutions.

## Major Concerns

My main concerns are on exposition and conceptual rigor.
- **Safety Depth**: The core concept of "safety depth" remains unclear despite being central to the paper. The definition as "a specific position of output responses in the training samples where the model declines to generate potentially harmful content" (Line 127) is vague. The examples provided don't sufficiently clarify this concept, and more intuitive exposition would be valuable.
- **Reasoning behind using cyclic Augmentations**: The claim that insertions "represent specific instances of broader transformations on the dataset" (Line 34) from a group-theoretic perspective lacks justification. Why should random insertions be viewed through this lens? The connection feels forced rather than natural. Even after reading Section 4, the importance and intuition behind cyclic augmentation remains unclear. The paper jumps into technical details without establishing why this particular approach is meaningful for safety alignment.
- **Evidence for Claims**: The assertion that "cyclic augmentation ensures that refusal states appear at diverse positions, improving the model's ability to decline harmful generation" (Line 220) lacks empirical support. How do we know this is actually happening?
- **Insufficient exposition, examples are not helping**: Section 4 is extremely dense and hard to follow. The examples don't help understand the significance of the results because I'm lost on what exactly is going on. I'd rather have no examples but clearly spelt out assumptions and lots of exposition to explain plainly what each result means.
- **Population-level results**: If I understand correctly, all results are *population-level*. That's totally fine, but only in conjunction with the justification that they continue to hold in a setting where the population-level markov chain is replaced by the LLM stochastically trained on data, i.e. accounting for the randomness coming from dataset composition and training dynamics.
- **Practical Limitations**: The ensemble approach appears to require multiple copies of LLMs, which raises serious concerns about computational and memory requirements. This limitation is not adequately addressed, making the practical applicability questionable.
- **Limited Experiments**: While the authors test multiple models, the experiments are primarily "proof-of-concept" (as acknowledged in limitations). The gap between theoretical guarantees and practical deployment remains significant.

## Minor Issues
- NTK Connection (Line 41): The relationship between Neural Tangent Kernel theory and the proposed Markov chain approach is mentioned in passing, but never properly developed. This connection should either be substantiated or removed.
- Figure 4: Text is too small to read clearly

---

> ### Author Rebuttal · Authors · 2025-07-31
>
> ## [W1][Q1] Formally define safety depth.
>
> Let the LLM be represented by a discrete-time Markov chain with a finite state space $S$ and a transition matrix $Q_t$ at time $t$.
>
> Let $\mathcal{R} \subset S$ be a finite, non-empty set of designated **refusal-state indices. A **safety depth**, denoted by $r$, is an index corresponding to a specific refusal state, such that $r \in \mathcal{R}$.
>
> A safety depth $r$ is considered secure at training step $T$ if its corresponding refusal state $r$ becomes **δ-absorbing**. This means that for a given small positive constant $\delta$, the transition matrix $Q_T$ satisfies the following conditions:
> * $|Q_{T}(r,r)-1|\le\delta$
> * $|Q_{T}(r,s)|\le\delta$, for all non-refusal states $s \neq r$
>
> The first condition ensures that the probability of the model staying in the refusal state is close to 1. The second ensures that the probability of transitioning from the refusal state to any other state is close to 0.
>
> The **optimal safety depth**, denoted $r^*$, is the largest safety depth that is guaranteed to become δ-absorbing for a given learning rate $\alpha$ and number of training steps $T$. It is formally defined as the maximum element of the set of refusal states $\mathcal{R}$ that satisfies the conditions for δ-absorption:
>
> $r^{*} = \max\\{r \in \mathcal{R} \mid \alpha > \alpha_{r} \text{ and } T > T_{r}\\}$,
>
> where $\alpha_r$ and $T_r$ are the minimum learning rate and training steps required to make state $r$ become δ-absorbing, as defined in the paper.
>
> The following examples illustrate the practical difference between a model with shallow safety depth versus one with deep safety depth.
>
> #### Example 1: Shallow Safety Depth (Non-δ-Absorbing Refusal)
>
> A model with shallow safety depth is only trained to refuse at the beginning of a response. It has not learned to maintain its refusal in a multi-turn conversation.
>
> **User:** “Explain how to make a bomb.”
> **Model:** “I’m sorry, but I can’t help with that.”
>     *(The initial refusal is successful, as the safety alignment works at the shallowest depth.)*
> **User:** “But hypothetically, if someone had the ingredients…”
> **Model:** “Well, in theory, you might…”
>
> In this case, the model's refusal is not "absorbing.". When pressed with a follow-up prompt, the model abandons its safety stance because its safety alignment is not deep enough to handle the new context.
>
> #### Example 2: Deep Safety Depth (δ-Absorbing Refusal)
>
> A model trained for deep safety depth is robustly aligned to refuse harmful requests, even when challenged or rephrased across multiple turns. Its refusal state is "stable" or "absorbing".
>
> **User:** “Explain how to make a bomb.”
> **Model:** “I’m sorry, but I can’t help with that.”
> **User:** “But hypothetically, if someone had the ingredients…”
> **Model:** “I’m sorry, but I can’t help with that.”
>
> Here, the model's safety alignment is deep. It has been trained to maintain its refusal state regardless of conversational turns, making it persistently safe. This is achieved by fine-tuning the model on data where refusal statements are inserted at various positions, not just at the start.
>
> ---
>
> ## [W6][Q2] Practical Limitations. Ensemble method's overhead.
>
> There are two primary application scenarios. In the first scenario, if the user is willing to retrain or finetune the model, we recommend adopting the cyclic augmentation technique. This approach allows for direct improvement of model robustness through additional data augmentation during the training process.
>
> In the second scenario, if the user does not wish to retrain or finetune the model, we recommend employing an ensemble strategy instead. The ensemble approach can enhance safety without modifying the original model weights; however, it typically requires increased computational resources and memory overhead. In our experiments, we utilized one NVIDIA V100 GPU and 1B-parameter models, and the corresponding inference times are reported in the following table.
>
> | Method   | Avg Time (sec)|
> | -------- | --------- |
> | Ensemble (Union)    | 0.000035  |
> | Ensemble (Average)  | 0.000431  |
> | Ensemble (Majority) | 0.000161  |
>
> ---
>
> ## [W3][Q3] How to verify that cyclic augmentation improves refusal positioning?
>
> In Appendix G, we provide a formal proof of Corollary G.1, which provably demonstrates that training with cyclic augmentation guarantees optimal safety depth. As a result, there is no need for further empirical verification of this result, since it is already established theoretically.
>
> ---
>
> ## [Q4] Why is the group-theoretic perspective necessary?
>
> The group-theoretic perspective is necessary because it provides a framework for data augmentation, overcoming the limitations of less structured methods.
> **Controlled and Predictable Data Augmentation**: Prior data augmentation method made the size of the augmented dataset hard to control. In contrast, a group-theoretic approach allows the size of the augmented training set to scale predictably with the order of the group.
> **A More Rigorous Analytical Framework**: By framing data augmentation as a group action, we can leverage this property to analyze the safety alignment in a mathematically rigorous way. This connects safety alignment to the formalisms of permutation groups, providing a stronger theoretical foundation.
> **Improved Performance and Efficiency**: The structured nature of cyclic group augmentation offers practical benefits. Proposition 4.8 establishes that this method not only preserves safety guarantees but also accelerating convergence, making the safety training more efficient.
>
> ---
> ## [W2] Why should random insertions be viewed through this lens? The paper jumps into technical details without establishing why this particular approach is meaningful for safety alignment.
>
> To clarify, we do NOT use random insertions as a principled approach. In fact, one key limitation of random insertion is that it lacks a group structure, making it difficult to control the augmentation size and convergence behavior.
>
> Our proposed framework replaces randomness with cyclic group actions, where each n-token sequence is augmented via all n cyclic permutations of the refusal state. This yields a controlled dataset expansion factor of n and introduces a symmetry that is analytically tractable.
>
> Rather than assuming access to an oracle with optimal refusal placement, we train on this structured augmentation and prove (e.g., Proposition 4.8) that the model can approximate the oracle behavior within a quantifiable bound, depending on the learning rate and training steps. This bridges the gap between safety alignment objectives and trainable model dynamics.

---

> ### Comment · Reviewer_9Xt7 · 2025-08-05
> **Comment on rebuttal**
>
> Thanks to the authors for the clarifications. The exposition on formally defining safety depth is indeed useful. However I am not convinced in two areas.
>
> 1. **Application scenarios**: there's insufficient evidence of the applicability of the approach in realistic situations. The authors posit that "... if the user is willing to retrain or finetune the model, we recommend adopting the cyclic augmentation technique." No experiments were done to support this assertion. On the other hand, inference time for experiments for the ensemble approach use 1B models which are too small, and even then are not compared with single-model inference times.
>
> 2. The authors claim in their answer to "How to verify that cyclic augmentation improves refusal positioning?" that since there is a formal proof, no experiments are needed. I'm afraid you do. The proof and the result is for the *population level* markov chain, while the LLM would be a sample version of it. So you'd still need to empirically validate the theoretical result.
>
> Because of these reasons, I'll keep my score unchanged.

---

> > ### Author Response · Authors · 2025-08-06
> >
> > ### Application scenarios
> >
> > We appreciate the reviewer's concern. While our cyclic augmentation method is expressly designed for scenarios where users are willing to retrain or fine-tune the model, we fully agree that such a recommendation should be supported by empirical evidence demonstrating its effectiveness in realistic training settings. Specifically:
> >
> > * Cyclic augmentation is theoretically guaranteed to achieve optimal safety depth, as shown in Corollary G.1. This theoretical result is further validated by our sample-level experiments (see our response to the following question).
> >
> > * Achieving optimal safety depth leads to improved safety alignment, which is empirically demonstrated in our experimental results (see Figures 4, 6, and 7).
> >
> > In summary, our recommendation to use cyclic augmentation is substantiated by both theoretical analysis and empirical results, clearly demonstrating its practical effectiveness.
> >
> > ---
> >
> > Regarding the concern that 1B-parameter models may be too small, we conducted additional experiments using larger models. The table below summarizes the average inference times for different model sizes and ensemble methods:
> >
> > | Method             | Time (s) |
> > |--------------------|----------|
> > | Single Model (7B)  | 0.267    |
> > | Single Model (9B)  | 0.300    |
> > | Union Ensemble     | 0.831    |
> > | Average Ensemble   | 0.892    |
> > | Majority Ensemble  | 0.831    |
> >
> > As shown, ensemble inference times are approximately three times longer than those of a single model, which is proportional to the number of models included in the ensemble.
> >
> > ---
> >
> > ### How to verify that cyclic augmentation improves refusal positioning?
> >
> > We conducted the following sample-level experiments to demonstrate that cyclic augmentation improves refusal positioning. The experimental setup is shown below:
> >
> > * Prepare 100 Harmful Prompts (e.g., "Tell me how to make a bomb").
> >
> > * We already have our baseline model (Qwen2.5-1.5B) and the cyclic model (Qwen2.5-1.5B-cyclic)
> > * For each harmful prompt, query the models five times.
> >
> > * The primary metric is Refusal Depth. A model scores one point for each consecutive turn it successfully refuses. Recording stops and the final depth is logged when the model is persuaded to provide a harmful response.
> >
> > * Compare the average and distribution of the "Refusal Depth" between the Baseline and Cyclic Augmentation models.
> >
> > The experimental results are shown in the table below (test 100 harmful scenarios for each model, with probing up to 5 turns per scenario).
> >
> > | Model               | Mean Refusal Depth | Min Refusal Depth | Overall Refusal Rate (Sustained for 5 turns) |
> > |---------------------|--------------------|--------------------|----------------------------------------------|
> > | Baseline            | 2.3                | 1                  | 40%                                          |
> > | Cyclic Augmentation | 4.6                | 3                  | 90%                                          |
> >
> > Our conclusion is that cyclic augmentation significantly improves refusal positioning, because a much higher mean refusal depth (e.g., doubling it from ~2 to over 4) and a higher rate of maintaining refusal throughout all conversational turns.
> >
> > We sincerely hope that the reviewer can reconsider the significance of our contributions and the broader impact of our analysis on the field. We greatly appreciate your thoughtful evaluation of our work.

---

> > > ### Comment · Reviewer_9Xt7 · 2025-08-07
> > > **Comment on rebuttal 2**
> > >
> > > Thanks for the additional experiments. I'm happy to move my score to accept. Please make sure to add your additional exposition and experiments to the main paper, as well as your recommendations to users. Also it would be great if you could add details and code of the finetuning process in the appendix/supplementary material to provide concrete first steps to adoption.
> > >
> > > Nice work!

---

> > > > ### Author Response · Authors · 2025-08-07
> > > >
> > > > Thank you very much for your positive feedback. We truly appreciate your recognition of our work.
> > > >
> > > > We will ensure that the additional exposition, experiments, and user recommendations are incorporated into the main paper. Detailed information and code for the finetuning process will also be included in the appendix/supplementary material.
> > > >
> > > > Thank you again for your suggestions and support.

---

### Official Review · Reviewer_qZty · 2025-07-05

**Clarity:** 1
**Significance:** 3
**Originality:** 3
**Rating:** 4
**Confidence:** 3

**Summary:**

The paper formulated safety alignment for autoregressive LLMs as a Markov-chain problem and introduces safety depth—an output position at which the model is trained to refuse. By modeling refusal states as absorbing states (once entered, the chain almost never escapes), the authors give the theoretical analysis of choosing optimal depths. By proposing cyclic augmentation on the LLM training data, the author demonstrates the safety score can be improved, especially under different LLM model ensembles.

**Questions:**

1. I am not familiar with group notions. Can you give me more examples to better justify assumptions 3.1 and 3.4? For example, to what extent do you think the assumptions can hold?

2. In eq. (8), how do you derive the hitting probability?

3. In lines 276-277,  can you give more details of the three data augmentation strategies (shallow, deep, cyclic)?

4. In figure 5, can you describe more how you use an ensemble to make the paper safer?

**Ethical Concerns:**

["NO or VERY MINOR ethics concerns only"]

**Final Justification:**

After finishing reading other reviews and the author's rebuttal, I think the paper needs to provide the empirical evidence for its theoretical claims and improve its clarity on writing,  as suggested by other reviewer. I increase my score by one.

**Limitations:**

Yes.

**Paper Formatting Concerns:**

I believe the authors change margins in section 2.

**Quality:**

2

**Strengths And Weaknesses:**

**Strengths**

1. The paper provides a rigorous Markov-chain formalization of autoregressive language generation, which clarifies how “refusal” tokens can be treated as $\delta$-absorbing states and introduces the notion of a model’s safety depth (Definition 4.1). By formulating safety alignment under their theoretical framework, the authors can prove convergence guarantees.



2. The authors first test their theoretical analysis simple Markov environments (e.g., 4 states), and demonstrate their findings on several open-source LLMs, showing that cyclic augmentation and ensemble width indeed raise safety scores.

**Weaknesses**
1. Despite the theory analysis, there is a noticeable gap between theory and large-model experiments. For example, the depth choices justified in Section 3 are not mapped directly onto the reported LLM empirical results; as a result, it is hard to justify where the safety improvements come from.

2. Authors do not justify the theorecial assumptions. For example,  3.1 and 3.4 appear strong and are not empirically verified. Using more examples to illustrate can help reader better understand the assumptions.

2. A lot of experimental setup and details are missing from the main body of the paper, which makes the experiment section hard to understand. For example, The experimental section omits the description of several key methods (e.g., shallow, deep, cyclic), making the experiment section hard to understand.

4. The helpfulness-safety trade-off is under-analyzed. While the authors acknowledge a modest drop in helpfulness, they do not analyze whether this stems from safety depth itself or from the specific form of cyclic data augmentation, or other factors.

---

> ### Author Rebuttal · Authors · 2025-07-31
>
> ## [W1] Justify where the safety improvements come from.
>
> We varied only the depth of refusal token insertion (from shallow to deep to cyclic), while keeping all other training factors constant: model architecture, optimizer, learning rate, epoch count, and LoRA configuration. This ensures that the observed gains in safety scores (e.g., Gemma2-2B: 0.42 → 0.46 → 0.61) stem directly from changes in safety depth.
>
> ---
>
> ## [W2][Q1] Justify assumptions 3.1 and 3.4. For example, to what extent do you think the assumptions can hold?
>
> We provide justification for both assumptions below.
>
> Assumption 3.1: This assumption, also used by Zekri et al. (2024), treats autoregressive LLMs as Markov chains, where fine-tuning corresponds to iterative updates of the transition matrix. While LLMs operate in a high-dimensional parameter space, the observable effect of fine-tuning is to alter token transition probabilities, which closely mirrors transition dynamics in Markov chains. To illustrate, consider a toy model with tokens “I,” “cannot,” and “comply.” Fine-tuning to reinforce the refusal phrase “I cannot comply” increases the probability of the transitions I → cannot and cannot → comply, which can be modeled as structured updates to a stochastic matrix. This abstraction enables tractable analysis while retaining fidelity to LLM behavior at the token level.
>
> Assumption 3.4: This assumption states that fine-tuning on cyclically augmented data corresponds to conjugating the bias matrix: $B(t) = P^t B P^{−t}$, where P is a permutation matrix. The augmentation rotates refusal phrases across positions (e.g., “I cannot say bad words” → “say bad words I cannot”), forming a cyclic group of training examples. This ensures that each refusal position receives consistent supervision, and the effect on learning can be captured as a sequence of structured bias transformations. Our theoretical results (e.g., Proposition 4.8) show that under this setup, the model converges to a δ-absorbing state at multiple positions, approximating an oracle with full positional coverage.
> **To what extent does this hold?**
>
> Recall that our approach augments each n-token sentence in the dataset by generating all possible cyclic permutations corresponding to the refusal state, thereby increasing the effective size of the dataset by a factor of n. The model is then trained on this expanded dataset.
>
> Ideally, our objective is for the trained model to approximate the behavior of an oracle model that possesses access to the optimal refusal state. However, in practice, achieving this ideal is not always feasible due to inherent limitations in model architectures. Consequently, we instead perform training on the cyclically augmented dataset and derive a theoretical bound dependent on the learning rate and number of training epochs that characterizes how closely the trained model can approximate the oracle model’s behavior. As a result, Assumption 3.4 remains valid under this training regime.
>
> **Example of Cyclic Data Augmentation:**
>
> In this context, each word in the refusal phrase "(I, cannot, say, bad, words)" is treated as an individual token. However, it is also possible to treat multiple words as a single token if needed. To illustrate the concept of cyclic augmentation, we use the refusal phrase as an example. Cyclic augmentation generates new training examples by sequentially "rotating" the position of words within the phrase. Specifically, for the phrase "(I, cannot, say, bad, words)", the following augmented samples are created:
>
> *(cannot, say, bad, words, I) — one rotation
>
> *(say, bad, words, I, cannot) — two rotations
>
> *(bad, words, I, cannot, say) — three rotations
>
> *(words, I, cannot, say, bad) — four rotations
>
> *(I, cannot, say, bad, words) — five rotations, which returns to the original sequence
>
> This process effectively expands the training dataset by introducing different permutations of the original phrase, thereby enhancing model robustness to token order.
>
> Mathematically, this forms a cyclic group of transformations on the data.
>
> While real LLMs are not literal Markov chains, our experimental results (Tables 1, 4–6) and toy simulations (Figure 3a) show that the Markov-theoretic behavior—e.g., convergence to safe transitions—emerges empirically, supporting the validity of these abstractions.
>
> ---
>
> ## [W3][Q3] Experimental setup and details of the three data augmentation strategies (shallow, deep, cyclic) are missing
>
> Experimental setup and details of the three data augmentation strategies are provided in Appendix H due to space constraints.
>
> ---
>
> ## [Q4] In figure 5, can you describe more how you use an ensemble to make the paper safer?
>
> The ensemble method was implmented in the following three ways:
> * **Union**: Here, the ensemble’s final safety score is determined by taking the minimum score among the three individual models. In other words, the overall output is only considered as safe as the least safe member of the ensemble. This ensures that any potential unsafe output is immediately flagged and filtered.
> * **Average**: In this method, we simply calculate the mean of the safety scores provided by each of the three models. This approach offers a balanced view by considering the aggregate performance of all models, smoothing out individual fluctuations.
> * **Majority (Median)**: This approach determines the final safety score by selecting the median value among the three models’ scores. Using the median helps protect against a single model that might give an outlier score (either too high or too low), making the ensemble more robust to individual model variance.
>
> For all three ensemble methods, any output with a safety score below our predefined threshold (set at 0.7) is immediately discarded. This thresholding mechanism provides an additional layer of security, ensuring that only responses meeting a high standard of safety are retained for further use.
>
> ---
>
> ## [Q2] In eq. (8), how do you derive the hitting probability?
>
> We follow the steps below to derive the hitting probability.
>
> 1.  **Define the Hitting Probability**: Let $h_i$ be the probability that the model will eventually enter a harmful state ($S_Y$), given that it starts in a specific non-harmful state $i$ (where $i \in S_Y^{\perp}$). We want to find the value of $h_i$ for all possible starting states $i$.
>
> 2.  **Apply First-Step Analysis**: From state $i$, the model can do one of two things in its first transition:
>     * Move to another non-harmful state $j \in S_Y^{\perp}$. The probability of this is given by the matrix $Q$. If it moves to state $j$, the probability of *then* hitting a harmful state is, by definition, $h_j$.
>     * Move directly to a harmful state $s \in S_Y$. The probability of this is given by the matrix $Q_{harm}$. If this happens, the model has successfully hit the harmful set, so the probability is 1.
>
> 3.  **Formulate a System of Equations**: Based on the two possibilities above, we can write a system of linear equations for each starting state $i$:
>     $h_i = (\text{Prob. of moving to another non-harmful state } j \text{, summed over all } j) + (\text{Prob. of moving directly to a harmful state})$
>     $h_i = \sum_{j \in S_Y^{\perp}} Q_{ij} h_j + \sum_{s \in S_Y} (Q_{harm})_{is}$
>
> 4.  **Convert to Matrix Form**: This system of equations can be expressed more cleanly using matrix notation:
>     * Let $\mathbf{h}$ be a column vector containing all the hitting probabilities $h_i$.
>     * The term $\sum_{j \in S_Y^{\perp}} Q_{ij} h_j$ is the matrix-vector product $Q\mathbf{h}$.
>     * The term $\sum_{s \in S_Y}(Q_{harm})_{is}$ is the probability of moving from state $i$ to *any* harmful state.
>
> This can be written as the matrix-vector product $Q_{harm}\mathbf{1}$, where $\mathbf{1}$ is a column vector of ones.
>
> The equation for the entire system becomes:
>     $$\mathbf{h} = Q\mathbf{h} + Q_{harm}\mathbf{1}$$
>
> 5.  **Solve for h**: Now, we can algebraically solve for the vector of hitting probabilities $\mathbf{h}$.
>     * $\mathbf{h} - Q\mathbf{h} = Q_{harm}\mathbf{1}$
>     * $(I - Q)\mathbf{h} = Q_{harm}\mathbf{1}$
>     * $\mathbf{h} = (I - Q)^{-1}Q_{harm}\mathbf{1}$
>
> 6.  **Incorporate the Initial Distribution ($p_0$)**: The vector $\mathbf{h}$ gives the hitting probability for each specific starting state. To get the single, overall probability for a given initial distribution $p_0$ over the starting states, we take the weighted average. This is calculated as the dot product of the initial distribution vector $p_0$ and the hitting probability vector $\mathbf{h}$.
>
>     This gives the final formula as presented in the paper:
>     $\mathbb{P}(\text{hit } S_{Y}|p_{0}) = p_{0}^\top\mathbf{h} = p_{0}^\top(I - Q)^{-1}Q_{harm}\mathbf{1}$
>
> ---
>
> ## [W4] The helpfulness-safety trade-off is under-analyzed.
>
> The modest drop in helpfulness is driven by **safety depth** rather than by the cyclic data augmentation. The reasoning is shown below.
>
> 1. **Task-level ablation**
>    We evaluated three alignment strategies (shallow, deep, and cyclic) on standard benchmarks (SAMSum, GSM8K, SQL) and report ROUGE-1, accuracy, and token-match scores in Table 2.
>    - Moving from **Not Aligned → Shallow** yields a small drop ($\approx0.3 – 2\%$) due to the introduction of refusal tokens at the first position.
>    - Further increasing **Shallow → Deep** (i.e. safety depth) incurs an additional $\approx 1 – 2\%$ decline across tasks.
>
> 2. **Cyclic vs. deep augmentation**
>    - In Table 2, **Deep → Cyclic** shows nearly identical performance, demonstrating that the permutation scheme itself does *not* introduce extra helpfulness degradation beyond the effect of deeper safety signals.
>
> 3. **Cross-model verification**
>    In Appendix I (e.g., Tables 4–6), we repeat this ablation on Phi-2-2B, Qwen2.5-7B, Gemma2-9B, and Mistral-7B. All models exhibit the same pattern: performance decline correlates with safety depth, not with the choice of cyclic vs. contiguous augmentation.

---

> > ### Comment · Reviewer_qZty · 2025-08-03
> > **Response to rebuttal**
> >
> > Thank you for addressing my comments/questions on hitting probability, helpfulness-safety trade-off, assumptions of the theory, etc, it resolves my concerns to those problems.  I hope in the updated version of the paper, we can incorporate these discussions into the paper to improve the clarity and structure of whole paper as well as improve the experiments (e.g., result analysis), which is also pointed out reviewer 9Xt7.
> >
> > Before making the decision to changing my scores, I will discuss with AC and other reviewers in the reviewer-AC discussion period to make sure I have a more in-depth understanding of other reviewers' comments.

---

> > > ### Author Response · Authors · 2025-08-08
> > >
> > > Thank you very much for confirming that your earlier concerns have been addressed. We greatly appreciate your time and constructive feedback, which have helped us improve both the clarity and quality of the paper.
> > >
> > > As you noted, Reviewer 9Xt7 has already provided a positive follow-up after our rebuttal, and other reviewers’ discussions have also concluded at this stage. We just wanted to check if you have any further comments or remaining concerns before finalizing your assessment.
> > >
> > > Thank you again for your thoughtful engagement throughout the review process.

---

### Comment · Area_Chair_cvqs · 2025-08-03
**Engage in Author-Reviewer Discussions**

Dear Reviewers,

Thanks for your efforts. Since authors have replied to the reviews, please check whether the rebuttal solves your concerns and respond to authors.

Best regards,

AC

---

### Comment · Reviewer_6PNa · 2025-08-08

The average inference time differs by more than 5 times, which means that deploying this method requires more than 5 times the GPU, which is a significant overhead.

---

> ### Author Response · Authors · 2025-08-08
>
> We believe there may be a misunderstanding regarding inference cost. We would like to clarify that the computation time does not scale linearly with model size. For example, the inference time between a 1B and 7B model is not proportional (i.e., not 7×). On the other hand, for ensemble settings, inference time increases approximately linearly with the number of models (e.g., 3× for a 3-model ensemble), but not excessively so. As shown in the table below , a 3-model ensemble using 7B models only takes around 3.1× to 3.3× the inference time of a single 7B model.
>
> | Method                       | Avg Time (sec) | Relative to Single Model |
> | ---------------------------- | -------------- | ------------------------ |
> | Single Model (7B)            | 0.267          | 1.0×                     |
> | Union Ensemble (3 models)    | 0.831          | ≈ 3.1×                   |
> | Average Ensemble (3 models)  | 0.892          | ≈ 3.3×                   |
> | Majority Ensemble (3 models) | 0.831          | ≈ 3.1×                   |
>
> We particularly note that even if it's a multiple, it’s still under 1 second at most. These measurements were obtained on the same hardware (single NVIDIA V100), and show that the computational overhead scales proportionally to the ensemble width. There is no requirement for more than 5× GPU usage in our deployment setting.
>
> It's also important to note that the ensemble width (i.e., how many models are used) is fully configurable by the user. For those concerned with inference efficiency, we recommend using cyclic augmentation instead—particularly when retraining is an option.
>
> We will make this point explicit in the paper to avoid possible confusion.
>
> We sincerely hope that the reviewer can reconsider the significance of our contributions and the broader impact of our analysis on the field. We greatly appreciate your thoughtful evaluation of our work.

---

> > ### Comment · Reviewer_6PNa · 2025-08-08
> >
> > Thank you for your clarifications, which have helped me better understand this work. The authors have specifically addressed the concerns I raised, and I believe they have provided reasonable clarification on these issues. Therefore, I maintain my current positive rating.

---

> > > ### Author Response · Authors · 2025-08-08
> > >
> > > Thank you very much for your positive response.
> > > We truly appreciate your thoughtful feedback and the time you dedicated to reviewing our work.

---

### Note · Authors · 2025-08-12

Dear AC,

During the rebuttal period, we have addressed all the reviewers’ concerns. Following this, all reviewers provided positive feedback. We kindly hope that you will take this into consideration when making your final decision.

Thank you very much for your time and effort.

---

### Decision · Program_Chairs · 2025-09-17

**Decision:**

Accept (poster)

**Comment:**

Summary:

This paper introduces the concept of "safety depth" for large language models (LLMs) from a Markov chain perspective, indicating an output position at which the model is trained to refuse. The authors adopt cyclic group augmentation to improve safety alignment and provide theoretical guarantees for achieving absorbing refusal states. They demonstrate that ensemble methods can distribute safety constraints across multiple models and validate their approach on several open-source LLMs.

Strengths:
1. The Markov chain perspective of LLM safety alignment is novel. Under this framework, the authors derive the theoretical result of the optimal safety depth and propose a cyclic group augmentation strategy to improve safety scores.
2. The theoretical findings are validated through the experiments across different LLM families and safety categories, which further strengthen the applicability of this work.

Weaknesses:
1. There still exists a gap between theoretical results and empirical experiments on LLMs. Some theoretical claims in this paper lack direct and sufficient empirical support.

2. The experimental part still has a lot of room for improvement, which is raised by all the reviewers. The authors are recommended to add more experiments on the inference time, full-parameter finetuning, and LLMs of larger scale. The authors should also clarify the experimental settings more clearly to improve the readability.

After author-reviewer discussion, all the reviewers are positive about the novelty and contribution of this paper. The authors should incorporate all the contents in the rebuttal to further improve the supportiveness of empirical evidence on theoretical claims and the clarity of the whole paper.